# Anterior tooth-use behaviors among early modern humans and Neandertals

Kristin L. Krueger[1]*, John C. Willman[2,3], Gregory J. Matthews[4], Jean-Jacques Hublin[5], Alejandro Pérez-Pérez[6]

**1** Department of Anthropology, Loyola University Chicago, Chicago, Illinois, United States of America, **2** Institut Català de Paleoecologia Humana i Evolució Social (IPHES), Tarragona, Spain, **3** Àrea de Prehistòria, Universitat Rovira i Virgili (URV), Tarragona, Spain, **4** Department of Mathematics and Statistics, Loyola University Chicago, Chicago, Illinois, United States of America, **5** Department of Human Evolution, Max Planck Institute for Evolutionary Anthropology, Leipzig, Germany, **6** Department of Evolutionary Biology, Ecology and Environmental Sciences, Universitat de Barcelona, Barcelona, Spain

\* kkrueger4@luc.edu

**Data Availability Statement:** All relevant data are within the manuscript and its Supporting Information files.

**Funding:** This study was funded by the National Science Foundation (BCS-0925818) to KLK. JCW

## Abstract

Early modern humans (EMH) are often touted as behaviorally advanced to Neandertals, with more sophisticated technologies, expanded resource exploitation, and more complex clothing production. However, recent analyses have indicated that Neandertals were more nuanced in their behavioral adaptations, with the production of the Châtelperronian techno-complex, the processing and cooking of plant foods, and differences in behavioral adaptations according to habitat. This study adds to this debate by addressing the behavioral strategies of EMH ($n = 30$) within the context of non-dietary anterior tooth-use behaviors to glean possible differences between them and their Neandertal ($n = 45$) counterparts. High-resolution casts of permanent anterior teeth were used to collect microwear textures of fossil and comparative bioarchaeological samples using a Sensofar white-light confocal profiler with a 100x objective lens. Labial surfaces were scanned, totaling a work envelope of 204 x 276 μm for each individual. The microwear textures were examined for post-mortem damage and uploaded to SSFA software packages for surface characterization. Statistical analyses were performed to examine differences in central tendencies and distributions of anisotropy and textural fill volume variables among the EMH sample itself by habitat, location, and time interval, and between the EMH and Neandertal samples by habitat and location. Descriptive statistics for the EMH sample were compared to seven bioarchaeological samples ($n = 156$) that utilized different tooth-use behaviors to better elucidate specific activities that may have been performed by EMH. Results show no significant differences between the means within the EMH sample by habitat, location, or time interval. Furthermore, there are no significant differences found here between EMH and Neandertals. Comparisons to the bioarchaeological samples suggest both fossil groups participated in clamping and grasping activities. These results indicate that EMH and Neandertals were similar in their non-dietary anterior tooth-use behaviors and provide additional evidence for overlapping behavioral strategies employed by these two hominins.

is supported by funding from the Marie Skłodowska-Curie Actions (H2020-MSCA-IF-2016 No. 749188), AGAUR (Ref. 2017SGR1040) and URV (Ref. 2017PFR-URV-B2-91) Projects, and MICINN/FEDER: PGC2018-093925-B-C32.

**Competing interests:** The authors have declared that no competing interests exist.

## Introduction

The concept of "behavioral ingenuity" has long been linked to narratives explaining both the evolutionary success of early modern humans and the eventual demise of the Neandertals [1–9]. This concept is often measured using some suite of archaeological or paleobiological criteria posited as markers of socioeconomic flexibility or complexity. For instance, Upper Paleolithic stone-tool technology, with an emphasis on blades and projectiles, is associated with early modern humans and seen as an upgrade from the Mousterian tradition [5, 10]. Dietary comparisons between early modern humans and Neandertals, including those from molar microwear [11], stable isotopes [12,13], paleoethnobotanical studies [14–17], faunal analyses [3, 18, 19], and food processing [20] frequently indicate that the former had greater dietary flexibility or accessed a broader subsistence base that included aquatic resources, fast and elusive small game, a greater variety of plant foods, and improved food storage and processing capabilities. Further studies suggest that early modern human clothing was more complex, fitted, and specialized, resulting in superior thermal protection during the cold oscillations of Marine Isotope Stage (MIS) 3 and beyond [21–23]. Although these analyses buttress common notions of early modern human ingenuity, recent studies suggest that Neandertal adaptation was more developed and nuanced than previously thought.

Several lines of evidence collectively identify Neandertal behaviors that are similar, or comparable, to the behaviors of penecontemporaneous early modern humans. The Châtelperronian technocomplex, associated with Neandertals, points to their ability to produce curved backed blades, bladelets, and bone tools [24–27], and projectile technology is also documented in Neandertal contexts [28, 29]. Dental calculus studies, which emphasize plant rather than animal foods, expand the range of dietary flexibility for Neandertals and suggest they were consuming not only cooked, but potentially medicinal plants as well [30–33]. Moreover, the overall evidence for plant exploitation visible in the archaeological record is similar between early modern humans and Neandertals, indicating the latter hominin possessed the ability to process those resources and had a complex division of labor for resource acquisition [29, 34–36]. Neandertals were also found to be adaptable in their anterior tooth-use behaviors, with habitat being a highly influential factor in the type of tooth-use behaviors employed [37]. Paramasticatory behaviors were not limited to anterior teeth, as "para-facets" identified on postcanine teeth of Neandertals and early modern humans were attributed to cultural activities, and not dietary behaviors [38]. Recent studies also confirm that Neandertals were capable of symbolic behavior in the form of cave art [39], use of body ornaments, marine shells and pigments [27, 40], and construction of elaborate structures deep within karstic systems [41].

Neandertals being capable of such complex behaviors blurs the dividing line between "us" and "them." Indeed, the mosaic morphology of archaic and anatomically modern humans found in many of the earliest modern human fossils suggests complex population dynamics in the Late Pleistocene [42–48]. The evidence from skeletal morphology has since been confirmed by aDNA evidence, with both nDNA and mtDNA analyses indicating multiple and earlier gene flow events, respectively, between early modern humans and Neandertals [49, 50].

This begs the following questions: what advantage did early modern humans have over Neandertals? What behavioral differences between these two hominins allowed us to proliferate and them to disappear? This study seeks to add to the debate using dental microwear texture analysis as a means to compare early modern human tooth-use behaviors with those of the Neandertals. Anterior tooth-use behaviors serve as a proxy for determining the degree to which Neandertal and early modern human groups relied on their anterior teeth and jaws for manipulative behaviors. Less intensive use of the teeth for such activities in EMH may suggest a different repertoire of behavioral strategies.

## Tooth-use behaviors in the Paleolithic

Neandertals are often associated with a particular collection of anterior tooth wear patterns, including labial rounding, labial scratches, and differential anterior-posterior occlusal wear, as they were documented on numerous individual fossils across time and space [51–70]. As a result, several hypotheses were put forth to explain the etiology of these wear patterns, including specialized chewing [52], coarse food and non-dietary behaviors [53], excessive mastication of abrasive foods [71–73], and different combinations of dietary and non-dietary behaviors [58, 74, 75]. The use of the anterior teeth for different types of non-dietary behaviors is now well-established, but the most common behavioral reconstruction centered on the so-called "stuff-and-cut" action. This posited that Neandertals were using their anterior dentition as a third hand to clamp down on meat or hide, and slicing it near their mouths with a stone tool [53–57, 76, 77].

This behavioral reconstruction of Neandertal tooth-use became conventional wisdom, even though variation in non-dietary anterior tooth-use behaviors were documented bioarchaeologically and ethnographically [78–86]. Analyses of anterior tooth-use among recent humans using dental microwear textures provide a comparative framework to document behaviors that extend well beyond the stereotypical "stuff-and-cut" action, including tool production and retouching, hide preparation, wood softening, and weaving tasks [87, 88]. Resulting microwear textures from the anterior teeth of a large sample of Neandertals (also used here) show significant variation in non-dietary anterior tooth-use behaviors, with habitat a prominent factor in distinguishing activities [37]. Specifically, individuals in more open habitats were participating in intense clamping and grasping behaviors, whereas those in more closed environments were engaged in a spectrum of non-dietary and dietary-only behaviors [37]. This ecogeographic patterning of anterior tooth-use behaviors is echoed by a similar pattern found in postcanine, dietary dental wear [89–91].

Early modern humans have largely been excluded from analyses of anterior tooth-use behaviors, with a few, notable exceptions. For instance, comparisons of Neandertal and early modern human anterior versus posterior occlusal macrowear gradients are well studied, and a pattern of greater anterior relative to posterior macrowear is common to both groups [58, 69, 70, 92– 94]. Some recent bioarchaeological groups and specific early modern humans exhibit greater anterior relative to posterior wear than many Neandertals [93]. However, Neandertal anterior teeth (incisors and canines) are larger on average than those of early modern humans [95], and more frequently exhibit mass-additive crown morphology (e.g., shoveling, *tuberculum dentale*, distal accessory ridges, etc. [96–98]). Therefore, the anterior teeth of Neandertals lose more volume per unit of occlusal wear than those of early modern humans, on average [58, 69, 70, 94]. Exploring anterior versus posterior dental macrowear gradients scaled to crown breadth in bivariate space highlights the distinctions between Neandertals and early modern humans anterior crown wear as it relates to differential anterior crown size; however, it is important to note that some samples demonstrate overlap at the 95% confidence interval of slope and y-intercept [58, 69, 70, 94]. Likewise, an analysis of dentin exposure by tooth, standardized to first molar wear, shows not only extensive variation in rates of anterior tooth wear, but also that some early modern and recent human groups exhibit far greater anterior dental wear than Neandertals [93]. The former analyses suggest few behavior differences between Neandertals and early modern humans, in that both groups engaged in anterior tooth-use typical of hunter-gatherers, but that tooth size dictates the functional "use-life" of an anterior tooth [69]. In contrast, the latter study suggests that there is no support for differences in anterior dental loading between Neandertals, early modern humans, and recent human groups given the overlapping or more extensive wear of anterior relative to first molar wear in the modern human groups [69].

Individual wear features, such as labial instrumental striations indicative of stuff-and-cut actions, are rarely examined among early modern humans. A recent study of the dental remains from Dolní Věstonice and Pavlov [94] showed that instrumental striations were ubiquitous on the well-preserved dentitions of these individuals. However, the striations were most frequently oriented vertically, and probably caused by downward scraping behaviors rather than the oblique cutting motions associated with most Neandertal labial striations [94]. Occlusal grooves [97] and lingual surface attrition of the maxillary anterior teeth were also found among the Pavlovian dentitions [94]. Taken together, the wear patterns exhibited by these early modern humans indicate extensive anterior tooth-use for clamping and grasping behaviors, probably related to hide preparation or similar activities [94, 99].

Although the data on early modern humans are limited, it seems that repetitive, manipulative behaviors associated with particular anterior dental wear patterns were not simply a Neandertal phenomenon [69]. Dental microwear texture analysis, with its standardized protocol and high repeatability, on a large sample of early modern humans and Neandertals presented here can further identify upon potential similarities or differences in manipulative behaviors among these Late Pleistocene human groups.

## Biomechanical versus comparative approach

Qualitative descriptions of Neandertal cranio-facial morphology and anterior tooth size and wear led researchers to hypothesize that the Neandertal face was adapted to high magnitude and/or repetitive loading of the anterior teeth [57, 100–102]. Referred to as the Anterior Dental Loading Hypothesis (ADLH), this theoretical approach posits that behavioral strategies involving the use of teeth-as-tools provided a selective force in Neandertal cranio-facial and dental evolution [100–107]. However, several specific morphological characteristics, including the retromolar space and posterior position of the zygomatic arch relative to the maxillary molars, sparked debate about the biomechanical efficiency and evolutionary significance of non-dietary anterior tooth-use in Neandertals [103, 104]. This led to several biomechanical modeling studies that indicated Neandertals were neither capable of nor efficient at high magnitude loading of the front teeth [93, 108–111], and Neandertal craniofacial evolution was the result of climate-based adaptations and/or neutral evolutionary forces, such as genetic drift [109, 112–119]. The challenge in using a biomechanical approach is it provides the potential for high-magnitude loading, but not direct evidence of it, leaving the question open as to what Neandertals actually did with their anterior teeth.

Direct analysis of anterior dental wear, such as dental microwear, macrowear, and different types of dental wear features (e.g., enamel chipping and instrumental striations), provide one means of directly assessing the behaviors that would (or would not) correspond to differential loading or use of the anterior dentition. These methods employ novel quantitative measurements, such as microscopic enamel textures [37, 87, 88], instrumental cutmark analyses [60, 63, 65–70, 120], and macrowear gradients [58, 69, 70, 92, 93] and Occlusal Fingerprint Analysis [38, 90, 121–125] to document Neandertal and early modern human behaviors using a comparative approach. These types of analyses often rely on a database of modern human samples with known or inferred dietary and tooth-use behaviors as a comparative benchmark for the fossils analyzed. There are also challenges with direct approaches, including sample size, sample composition, and assuming behavior in the ethnographic present is similar to that found in the Pleistocene [69]; however, these types of analyses have offered a fresh perspective on anterior tooth-use behaviors, including differences in Neandertal wear patterns driven largely by habitat [37], similar behaviors between Neandertals and Late Pleistocene humans [69, 70], and evidence for mixed-diet and cultural behaviors on posterior teeth [125]. As such,

this study utilizes a comparative approach, and, in an effort to mitigate the challenges mentioned above, we employ a robust comparative framework with sizable samples and varied dietary and behavioral repertoires, and quantitative data to support our conclusions.

## Materials and methods

### Fossil and comparative samples

The fossil sample is comprised of early modern humans ($n$ = 30) predominantly from Marine Isotope Stage (MIS) 3–2; however, those from Qafzeh and Skhūl are dated to MIS 5. These individuals are from 13 sites located across Europe and Israel (Table 1). The Neandertal sample ($n$ = 45) ranges in date from MIS 7–3 and spans across Western Eurasia (Table 2). The modern human comparative sample ($n$ = 156) consists of seven groups that range in time from 5000–100 years BP (Table 3). These individuals lived in a wide variety of environments, exploited various resources, and differed in non-dietary anterior tooth-use behaviors [37, 86, 87].

The early modern human sample is evaluated using three factors: habitat, location, and time interval [37]. The two habitat categories are based on vegetation cover, and include "open" and "mixed," and are similar to those used in molar microwear texture analyses [11, 89]. "Open" habitats are those that typically have less than 15% arboreal pollen, if palynology is available, and/or show a majority of open habitat-adapted fauna (e.g. *Rangifer*, *Equus*). "Mixed" habitats are those that contain a variety of landscapes, including the forest-steppe environments of Dolní Věstonice and Pavlov and the woodland, grassland, marsh, desert, and aquatic habitats of Ohalo II. Palynology, when available, falls between 20–60% and/or includes fauna indicative of a variety of landscapes (e.g. *Rangifer*, *Cervus*, *Equus*, *Sus*, etc.). Table 2 includes Neandertals found in "covered" habitats, which indicates over 60% arboreal pollen and forest-dwelling fauna. Temperature is not taken into consideration because while the "open" group is associated with colder temperatures, the "mixed" group encompasses sites that would have differed dramatically in temperature. The goal here is to discern adaptations according to vegetation availability, and not temperature.

Location is divided into three categories, Western Europe, Central Europe, and Southwest Asia. The time interval category is based on MIS intervals, which includes 5, 3, and 2. We

**Table 1. Summary of the early modern human sample used in this study.**

| Country | Site | $n$ | Habitat | Location | MIS |
|---|---|---|---|---|---|
| Czech Republic | Dolní Věstonice | 4 | Mixed | Central Europe | 3 |
| | Pavlov I | 4 | Mixed | Central Europe | 3 |
| France | Brassempouy | 2 | Open | Western Europe | 3 |
| | Farincourt | 1 | Open | Western Europe | 2 |
| | Isturitz | 1 | Mixed | Western Europe | 2 |
| | Lachaud | 2 | Open | Western Europe | 2 |
| | Les Rois | 5 | Open | Western Europe | 3 |
| | Rond-du-Barry | 1 | Open | Western Europe | 2 |
| | Saint-Germain-la-Rivière | 1 | Open | Western Europe | 2 |
| Italy | Grotte des Enfants | 1 | Open | Western Europe | 3 |
| Israel | Ohalo II | 1 | Mixed | Southwest Asia | 2 |
| | Qafzeh | 4 | Mixed | Southwest Asia | 5 |
| | Skhūl | 3 | Mixed | Southwest Asia | 5 |
| **TOTAL** | | **30** | | | |

See S1 File for more detailed information about each specimen.

recognize the challenges in grouping samples chronologically by broad MIS designations, but these designations correspond to group divisions of biological and archaeological relevance. For instance, the MIS 5 group corresponds to modern humans from Skhūl and Qafzeh with Middle Paleolithic material culture, the MIS 3 group largely corresponds to early Upper Paleolithic modern humans, and the MIS 2 group largely corresponds to the post-Last Glacial Maximum humans with Late Upper Paleolithic/Epipaleolithic material culture.

The Neandertal comparative sample ($n = 45$) consists of individuals that span their geographic and temporal ranges and come from "open," "mixed," and "closed" habitats (Table 2; [37]). As stated above, only those Neandertals from the "open" and "mixed" categories ($n = 25$) are used in the habitat comparisons. The location designations are the same as those described for the early modern human sample, with the entire Neandertal sample used in analysis ($n = 45$). The early modern humans and Neandertals are not compared by time, as the Neandertal sample required a broader chronological grouping, "early" (MIS 7–5) and "late" (MIS 4–3), due to limitations in dating techniques and their ranges [37].

Grouping fossil material is a challenge, as there are inconsistent data on excavation histories, stratigraphic context, environmental reconstructions, dating techniques. We have attempted to standardize these datasets as much as possible, as shown in the S1 File (and SOM in [37]); however, these limitations resulted in broad categories. We recognize that other researchers may use different groupings [90, 126]. All data are available for continued analysis, and can be found in the S1 File (and SOM in [37]).

**Table 2. Summary of the Neandertal sample used in this study.**

| Country | Site | n | Habitat | Location | Chronology |
|---|---|---|---|---|---|
| Croatia | Krapina | 10 | Closed | Central | Early |
| | Vindija | 4 | Mixed | Central | Late |
| Czech Republic | Kůlna | 1 | Mixed | Central | Late |
| | Ochoz | 1 | Mixed | Central | Late |
| France | Arcy-sur-Cure, Grotte de l'Hyène | 2 | Open | Western | Late |
| | Biache-Saint-Vaast | 1 | Closed | Western | Early |
| | Combe Grenal | 1 | Open | Western | Late |
| | La Chaise, Abri Suard | 1 | Open | Western | Early |
| | La Chaise, Abri Bougeois-Delaunay | 2 | Open | Western | Early |
| | La Ferrassie | 2 | Mixed | Western | Late |
| | La Quina | 1 | Open | Western | Late |
| | Le Moustier | 1 | Open | Western | Late |
| | Le Petit-Puymoyen | 1 | Open | Western | Late |
| | Les Pradelles (Marillac) | 1 | Open | Western | Late |
| | Las Pélénos (Monsempron) | 1 | n/a | Western | Late |
| | Moula Guercy | 3 | Closed | Western | Early |
| | Saint-Césaire | 1 | Mixed | Western | Late |
| Great Britain | Pontnewydd | 1 | Mixed | Western | Early |
| Hungary | Subalyuk | 1 | Open | Central | Late |
| Spain | Zafarraya | 3 | Closed | Western | Late |
| Iraq | Shanidar | 1 | Mixed | SW Asia | Late |
| Israel | Amud | 2 | Mixed | SW Asia | Late |
| | Kebara | 1 | Mixed | SW Asia | Late |
| | Tabūn | 2 | Closed | SW Asia | Early |
| **TOTAL** | | **45** | | | |

See [37] for information on how each site was categorized.

**Table 3. Summary of the modern human comparative samples used in this study.**

| Group | Location | n | Date (yrs BP) | Environment | Non-dietary tooth use? |
|---|---|---|---|---|---|
| Andamanese | Andaman Islands | 15 | 150 | Tropical, monsoon | Yes, tool retouching, production, "stuff and cut" practices |
| Arikara | Mobridge, South Dakota | 18 | 400–300 | Grassland | No |
| Chumash | Northern Channel Islands, CA | 19 | 5000–4000 | Cool Mediterranean | No |
| Sadlermiut | Northwest Hudson Bay, Canada | 27 | 950–100 | Polar arctic | Yes, intense clamping and grasping |
| Tigara | Point Hope, AK | 34 | 750–250 | Arctic, arid | Yes, some clamping and grasping, sinew thread production |
| Coast Tsimshian | Prince Rupert Harbour, Canada | 25 | 4000–700 | Oceanic, temperate | Yes, weaving tasks |
| Puye Pueblo | Pajarito Plateau, NM | 18 | 1100–330 | Desert | No |
| **TOTAL** | | **156** | | | |

See [37] for more detailed information on each group.

The modern human comparative sample (*n* = 156) consists of seven groups including the Andaman Islanders (*n* = 15), located in the Bay of Bengal, and Arikara (*n* = 18), Chumash (*n* = 19), Nunavut Territory Sadlermiut (*n* = 27), Point Hope Tigara (*n* = 34), Prince Rupert Harbour Coast Tsimshian (*n* = 25), and Puye Pueblo (*n* = 18) indigenous North American populations. These groups lived in a wide range of geographic locations, inhabited different environmental conditions, and accessed various plant and animal resources (Table 3). They also participated in a variety of non-dietary anterior tooth-use behaviors [37, 87, 88]. Ethnographic evidence indicated the Andaman Islanders used their anterior teeth for tool retouching and stuff-and-cut actions [78, 79, 127], whereas the Point Hope Tigara engaged in some clamping and grasping behaviors for hide and sinew production [84, 128–130]. The Nunavut Territory Sadlermiut participated in an intense regimen of clamping and grasping for hide production [131–133] and the Prince Rupert Harbour Coast Tsimshian softened plant fibers for weaving tasks [82]. These behaviors were inferred from datasets independent of microwear, such as indigenous oral histories, archaeological remains, and other dental analyses, including macrowear and chipping. There is no evidence that the Arikara, Chumash, or Puye Pueblo participated in non-dietary anterior tooth-use behaviors.

## Dental microwear texture analysis

High-resolution casts of the early modern human, comparative Neandertal, and recent modern human samples were used in this analysis. As statistical analyses indicate that microwear textures do not differ significantly across anterior tooth types [37], all anterior tooth types were included for the fossil samples in order to expand the sample size to its greatest capacity. Only maxillary central incisors of the recent modern human samples were used here because of increased preservation and availability.

The labial surface of the analyzed tooth was cleaned gently with acetone and cotton swabs prior to molding. The molding and casting materials used were President Jet regular body (Coltène-Whaledent) and Epotek 301 epoxy (Epoxy Technologies), respectively. Antemortem microwear was scanned on the labial surface, nearest the incisal edge, using a Sensofar Plμ white-light confocal profiler (Solarius Development Inc., Sunnyvale, CA). All specimens were scanned using the same confocal profiler ("Connie") at the University of Arkansas to avoid inter-microscope variation [134].

Four adjacent scans of the labial surface were taken using a 100x objective lens; this created a total sampling area of 204x276 μm [135]. The scans were examined for surface defects, such as taphonomic damage, using Solarmap Universal software (Solarius Development Inc., Sunnyvale, CA). If such defects existed, they were deleted before being characterized using

Toothfrax and SFrax scale-sensitive fractal analysis software (Surfact, www.surfract.com). Anisotropy (*epLsar*) and textural fill volume (*Tfv*) are the two texture variables considered here; their mathematical descriptions are described in Scott et al. [135].

These two texture variables in particular have been useful for distinguishing dietary from non-dietary behavioral regimes. Anisotropy (*epLsar*), or texture orientation, is elevated in groups who use their anterior dentition for incising food items only, and lower in those participating in non-dietary behaviors [37, 87, 88, 136]. The functional implication is that food (and/or adherent abrasives) are being dragged apically on the labial surface, creating parallel textures, which results in higher anisotropy values. On the other hand, using the anterior teeth in a variety of ways, including non-dietary behaviors, results in a lack of texture orientation on the labial surface [37, 87, 88, 136]. Textural fill volume (*Tfv*) is an indicator of bite force, with heavier or lighter bite force resulting in elevated or lowered textural fill volume values, respectively [37, 87, 88, 136]. For example, intense clamping and grasping with the anterior dentition would require a heavy bite force to maintain the material between the teeth. This would create large, deep textures, which results in high textural fill volume values [37, 87, 88, 136].

## Statistical analyses

There were two main goals in this study. The first was to examine only the early modern human dataset (*n* = 30) for significant variation in microwear textures (*epLsar* and *Tfv*) by habitat, site location, and time. The second was to compare central tendencies and distributions of *epLsar* and *Tfv* between the early modern human and Neandertal samples. All tests were completed using R statistical software; specific information for each goal can be found below [137].

First, the early modern human sample was examined for significant variation in anisotropy (*epLsar*) and textural fill volume (*Tfv*) by habitat, location, and time. For each combination of texture variables (i.e. *epLsar* and *Tfv*) and categorical predictor (i.e. habitat, location, and time),—six combinations in total—a one-way ANOVA was performed to look for significant differences in the means of *epLsar* and *Tfv* between the groups.

Second, the early modern human sample was compared with that of the Neandertals to determine if differences exist between these two hominins. A one-way ANOVA was completed first to compare the mean anisotropy and textural fill volume values between early modern humans (*n* = 30) and Neandertals (*n* = 45) as a whole. Next, a two-way ANOVA was conducted to look for differences between early modern humans (*n* = 30) and Neandertals (*n* = 45 for location, *n* = 25 for habitat) while controlling for location and habitat. As early modern humans in this dataset are not found in closed habitats, the closed-habitat Neandertals were removed from the habitat analysis, resulting in the lower sample size.

It is important to note that there were some data points in the Neandertal sample for both anisotropy and textural fill volume that exhibited high statistical influence. To reduce the impact of these data points on the parameter estimates, a robust regression using iteratively re-weighted least squares (IRLS), was performed; however, results were largely the same when compared to results obtained using traditional ANOVA analysis. In addition to looking for differences in central tendencies, Kolmogorov-Smirnov tests were performed to test for differences in the distributions of *epLsar* and *Tfv* between the two hominin groups. All R code used for statistical analyses can be found in the S2 File.

## Results

Visual and numerical results are found in Figs 1 and 2 and Tables 4–12, respectively. The stark uniformity of *epLsar* and *Tfv* values within the entire early modern human sample (Tables 4 and 5) is reflected in the lack of significant differences in central tendencies and distribution

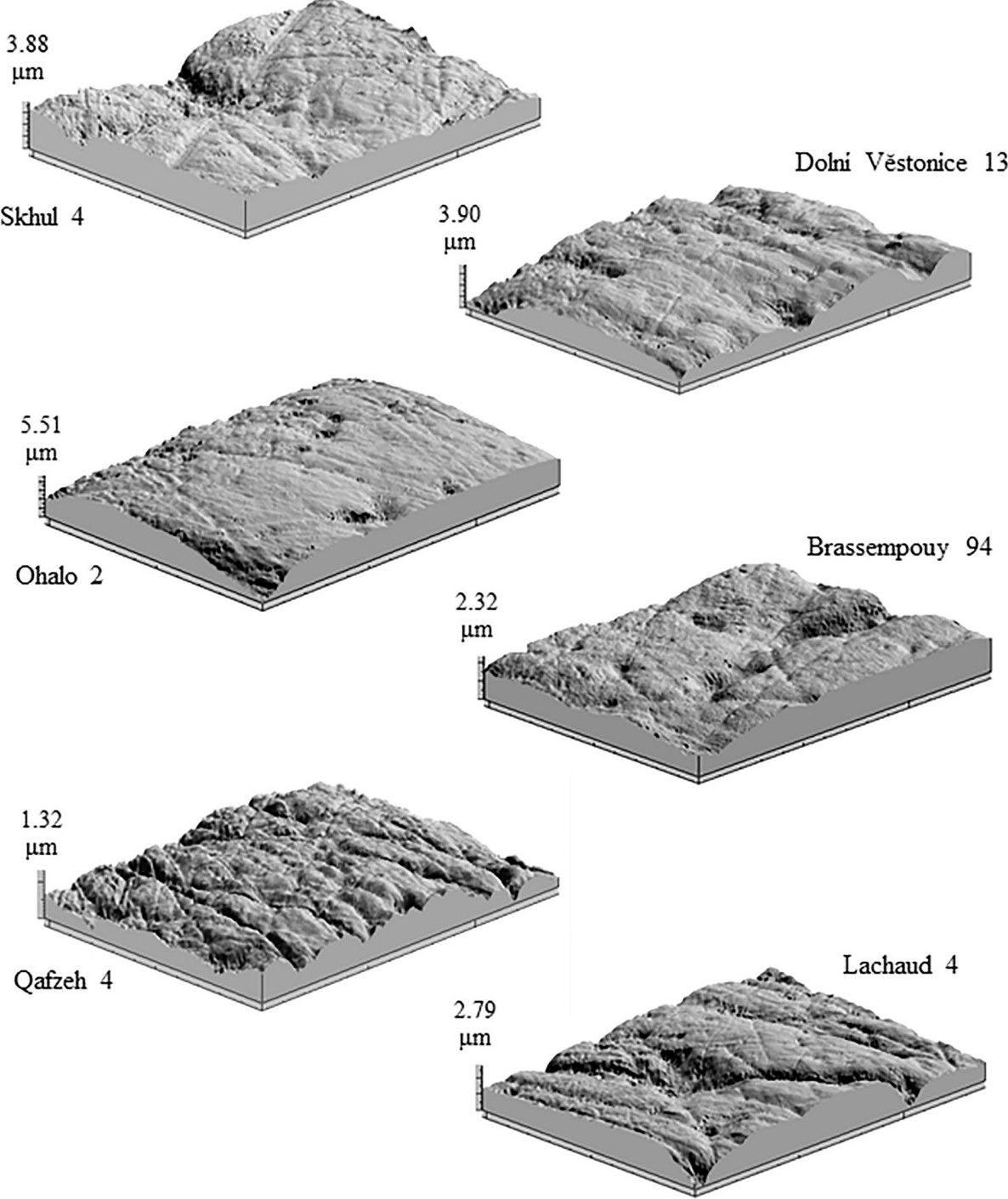

**Fig 1. Three-dimensional point clouds of early modern human anterior dental microwear surfaces.** Each image measures 102x138 μm; total area analyzed was 204x276 μm.

by habitat, location, or time (Table 6). Simply put, the early modern human sample had very similar anisotropy and textural fill volume values regardless of the factors considered here (see S1 File).

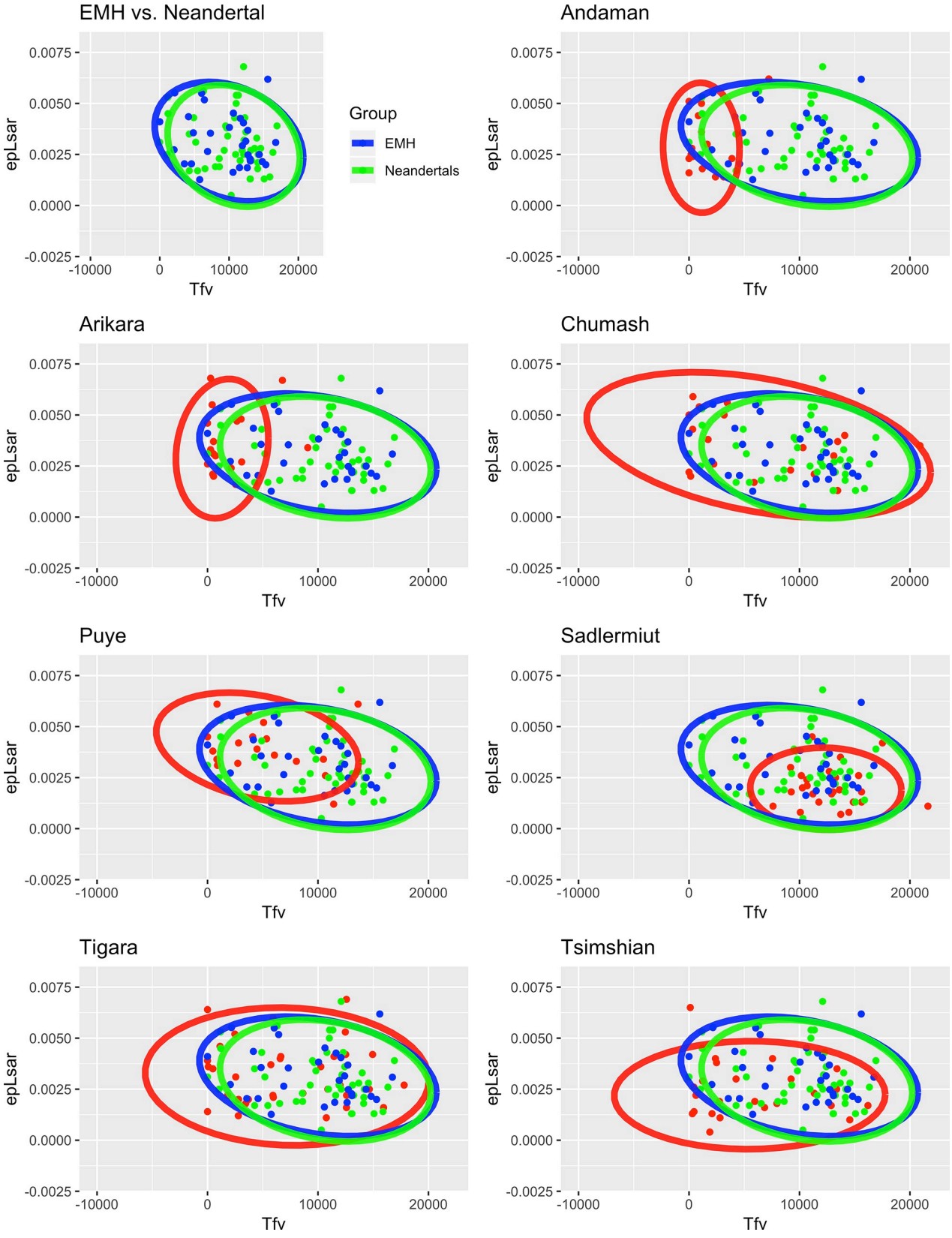

**Fig 2. Data plots with 95% confidence interval ellipses for Neandertals, early modern humans, and bioarchaeological comparative samples.** X-axis and Y-axis displays *epLsar* and *Tfv* values, respectively. Upper left: Neandertals (green) and early modern humans (blue) only, other plots show each individual bioarchaeological comparative group in red (labeled at the top), with Neandertals (green) and early modern humans (blue).

Neandertals vs. early modern humans (A); for open-habitat Neandertals vs. open-habitat early modern humans (B); for Western Europe, Central Europe, and Southwest Asia Neandertals vs. their EMH counterparts (C).

The second analysis examined *epLsar* and *Tfv* differences between the early modern human and Neandertal samples without considering any other factors. Again, no significant results were found between these two hominins in either central tendencies or distribution (Tables 7

**Table 4. Descriptive statistics for fossil and modern samples used in this study.**

| Sample | *n* | *epLsar* | *Tfv* |
|---|---|---|---|
| *Early modern humans* | 30 | | |
| Mean | | 0.0032 | 9520.10 |
| Median | | 0.0030 | 11071.43 |
| Std. Deviation | | 0.0013 | 4620.41 |
| *Neandertals* | 45 | | |
| Mean | | 0.0031 | 10117.77 |
| Median | | 0.0027 | 11041.15 |
| Std. Deviation | | 0.0014 | 4346.64 |
| *Andamanese* | 15 | | |
| Mean | | 0.0031 | 1559.29 |
| Median | | 0.0025 | 1127.43 |
| SD | | 0.0015 | 1965.24 |
| *Arikara* | 18 | | |
| Mean | | 0.0036 | 1897.76 |
| Median | | 0.0032 | 634.31 |
| SD | | 0.0016 | 2466.36 |
| *Chumash* | 19 | | |
| Mean | | 0.0035 | 6532.50 |
| Median | | 0.0035 | 3465.40 |
| SD | | 0.0014 | 6429.48 |
| *Nunavut Sadlermiut* | 27 | | |
| Mean | | 0.0020 | 12449.27 |
| Median | | 0.0018 | 12905.65 |
| SD | | 0.0010 | 3464.04 |
| *Tigara* | 34 | | |
| Mean | | 0.0032 | 7296.02 |
| Median | | 0.0029 | 6269.71 |
| SD | | 0.0015 | 5391.20 |
| *Prince Rupert Tsimshian* | 25 | | |
| Mean | | 0.0024 | 5766.64 |
| Median | | 0.0019 | 3079.71 |
| SD | | 0.0013 | 5196.40 |
| *Puye Pueblo* | 18 | | |
| Mean | | 0.0040 | 5093.03 |
| Median | | 0.0039 | 4284.68 |
| SD | | 0.0012 | 4183.08 |

**Table 5. Descriptive statistics of the early modern human (*n* = 30) and Neandertal (*n* = 45) comparative samples by habitat, site location, and time interval.**

| A. Habitat: | Early modern humans | | Neandertals | |
| --- | --- | --- | --- | --- |
| | *epLsar* | *Tfv* | *epLsar* | *Tfv* |
| **Closed** | *n/a* | | *n = 19* | |
| Mean | - | - | 0.0036 | 8380.87 |
| Median | - | - | 0.0038 | 9504.07 |
| SD | - | - | 0.0013 | 4020.37 |
| **Mixed** | *n = 17* | | *n = 14* | |
| Mean | 0.0031 | 9705.57 | 0.0031 | 10893.91 |
| Median | 0.0026 | 10602.81 | 0.0028 | 12603.01 |
| SD | 0.0004 | 1129.52 | 0.0015 | 5180.67 |
| **Open** | *n = 13* | | *n = 11* | |
| Mean | 0.0035 | 9069.41 | 0.0022 | 12204.74 |
| Median | 0.0035 | 11514.50 | 0.0021 | 12423.39 |
| SD | 0.0012 | 5011.23 | 0.0009 | 2776.93 |
| **B. Site Location:** | **Early modern humans** | | **Neandertals** | |
| **Western Europe** | *n = 14* | | *n = 22* | |
| Mean | 0.0034 | 9308.15 | 0.0028 | 11424.84 |
| Median | 0.0033 | 11572.99 | 0.0025 | 11748.97 |
| SD | 0.0011 | 4896.80 | 0.0014 | 3027.58 |
| **Central Europe** | *n = 8* | | *n = 17* | |
| Mean | 0.0032 | 9278.37 | 0.0033 | 8671.28 |
| Median | 0.0024 | 8738.40 | 0.0031 | 9661.46 |
| SD | 0.0017 | 4227.20 | 0.0014 | 4435.20 |
| **Southwest Asia** | *n = 8* | | *n = 6* | |
| Mean | 0.0029 | 10132.76 | 0.0034 | 9423.57 |
| Median | 0.0027 | 11275.56 | 0.0035 | 12660.30 |
| SD | 0.0011 | 5045.39 | 0.0014 | 7043.15 |
| **C. Time Interval:** | **Early modern humans** | | **Neandertals** | |
| **MIS 2** | *n = 7* | | **Early *n = 20*** | |
| Mean | 0.0032 | 9026.20 | 0.0031 | 9099.21 |
| Median | 0.0025 | 11514.50 | 0.0027 | 10234.04 |
| SD | 0.0015 | 4891.06 | 0.0011 | 4255.69 |
| **MIS 3** | *n = 16* | | **Late *n = 25*** | |
| Mean | 0.0033 | 9732.07 | 0.0030 | 10932.61 |
| Median | 0.0033 | 11104.38 | 0.0028 | 12094.76 |
| SD | 0.0013 | 4584.93 | 0.0016 | 4329.39 |
| **MIS 5** | *n = 7* | | | |
| Mean | 0.0030 | 9529.51 | | |
| Median | 0.0029 | 10628.36 | | |
| SD | 0.0012 | 5128.55 | | |

The time interval categories are different due to dating constraints within the Neandertal sample.

and 12A). When controlling for habitat and location, once again, there were no significant differences found between the early modern humans and Neandertals (Tables 8–11 and 12B). When visualized, the overall overlap in anisotropy and textural fill volume values between both hominin groups is remarkable (Fig 2). This overlap continues to be prevalent regardless of habitat type and location (Fig 2). The stark uniformity of dental microwear textures between

**Table 6. Results of the one-way ANOVAs for *epLsar* (A) and *Tfv* (B) within the early modern human sample only (*n* = 30).**

| A. *epLsar* | Estimate | Standard error | *p* value |
|---|---|---|---|
| (Intercept) | 0.0030682 | 0.0003160 | 1.84 e-10 |
| Open (habitat) | 0.0003880 | 0.0004801 | 0.426 |
| (Intercept) | 0.0031986 | 0.0004681 | 2.43 e-07 |
| Southwest Asia (location) | -0.0002711 | 0.0006619 | 0.685 |
| Western Europe (location) | 0.0002357 | 0.0005867 | 0.691 |
| (Intercept) | 3.246 e-03 | 5.042 e-04 | 6.74 e-07 |
| MIS 3 (time) | 9.491 e-05 | 6.045e-04 | 0.876 |
| MIS 5 (time) | -2.562 e-04 | 7.130e-04 | 0.722 |
| B. *Tfv* | Estimate | Standard error | *p* value |
| (Intercept) | 9864.8 | 1136.1 | 1.97 e-09 |
| Open (habitat) | -795.3 | 1725.9 | 0.648 |
| (Intercept) | 9278.37 | 1687.37 | 8 e-06 |
| Southwest Asia (location) | 854.39 | 2386.30 | 0.723 |
| Western Europe (location) | 29.77 | 2115.23 | 0.989 |
| (Intercept) | 9026.2 | 1806.3 | 3.07 e-05 |
| MIS 3 (time) | 705.9 | 2165.7 | 0.747 |
| MIS 5 (time) | 503.3 | 2554.5 | 0.845 |

this sample of Neandertals and early modern humans allows us to make inferences about their similar anterior tooth-use strategies and provides us with informed ideas concerning their overlapping manipulative behaviors.

Lastly, the early modern human sample shares texture values most similar to those of the Point Hope Tigara (Table 4, Fig 2). The anisotropy mean values are identical, and within the range of non-dietary anterior tooth-use behaviors. The textural fill volume values are similar, with the fossil sample showing an elevated value to that of the Tigara, but a lower mean value than that of the Nunavut Territory Sadlermiut. These comparisons offer the opportunity to possibly distinguish specific behaviors employed by the early modern human sample.

## Discussion and conclusion

### Early modern human sample

As a whole, the early modern human sample reflects texture values indicative of non-dietary anterior tooth-use behaviors that required a heavy loading regime (Table 4). Specifically, the anisotropy mean and median values indicate a lack of texture orientation, suggesting non-dietary behaviors. The textural fill volume mean and median values signify large, deep textures created by heavy loading regimes. Both mean texture values of the early modern human

**Table 7. Results of the one-way ANOVAs for *epLsar* (top) and *Tfv* (bottom) between Neandertals (*n* = 45) and early modern humans (*n* = 30).**

| *epLsar* | Estimate | Standard error | *p* value |
|---|---|---|---|
| (Intercept) | 0.0032363 | 0.0002448 | <2 e-16 |
| Neandertals (type) | -0.0001830 | 0.0003161 | 0.564 |
| *Tfv* | | | |
| (Intercept) | 9520.1 | 813.8 | <2 e-16 |
| Neandertals (type) | 597.7 | 1050.6 | 0.571 |

**Table 8. Results of the two-way ANOVA (A), robust regression (B), and 95% confidence intervals (C) for mean *epLsar* given habitat (open and mixed) and hominin type (Neandertal and early modern human).**

| A. | Estimate | Standard error | *p* value |
|---|---|---|---|
| (Intercept) | 3.068 e-03 | 3.120 e-04 | 2.27 e-13 |
| Open (habitat) | 3.88 e-04 | 4.739 e-04 | 0.4168 |
| Neandertals (type) | -3.92 e-06 | 4.642 e-04 | 0.9933 |
| Open-Neandertals | -1.26 e-03 | 7.023 e-04 | 0.0784 |
| B. | Regression co-efficient estimate | Standard error | *p* value |
| (Intercept) | 0.0029 | 0.0003 | 0.0000 |
| Open (habitat) | 0.0005 | 0.0005 | 0.2857 |
| Neandertals (type) | -0.0001 | 0.0004 | 0.8563 |
| Open-Neandertals | -0.0011 | 0.0007 | 0.0984 |
| C. | Fit | Lower | Upper |
| Neandertals (open) | 0.0022 | 0.0014 | 0.0030 |
| Neandertals (mixed) | 0.0031 | 0.0024 | 0.0038 |
| EMH (open) | 0.0035 | 0.0027 | 0.0042 |
| EMH (mixed) | 0.0031 | 0.0024 | 0.0037 |

sample are nearly identical to those of the Neandertals and closely align to those of the Point Hope Tigara modern human comparative sample (Table 4, Fig 2). These similarities indicate that overall, the early modern humans in this sample participated in tooth-use behaviors similar to those of the Neandertals, and those specific behaviors may be most akin to those employed by the Point Hope Tigara.

Examining the early modern human sample as a whole tells only part of the story, and it can be analyzed in finer detail by examining it by habitat type, location, and time interval to try to discern possible differences by these factors (Table 5). When this is done, interestingly, the story remains largely the same. The early modern humans show homogenous mean values in both texture variables regardless of habitat type, location, or time interval; this accounts for the lack of significant statistical differences (Table 6). Once again, these mean values are most similar to the Neandertal and Point Hope Tigara samples (Tables 4 and 5).

**Table 9. Results of the two-way ANOVA (A), robust regression (B), and 95% confidence intervals (C) for mean *Tfv* given habitat (open and mixed) and hominin type (Neandertal and early modern human).**

| A. | Estimate | Standard error | *p* value |
|---|---|---|---|
| (Intercept) | 9864.8 | 1095.5 | 4.05 e-12 |
| Open (habitat) | -795.3 | 1664.2 | 0.635 |
| Neandertals (type) | 1029.2 | 1630.1 | 0.531 |
| Open-Neandertals | 2106.2 | 2466.0 | 0.397 |
| B. | Regression co-efficient estimate | Standard error | *p* value |
| (Intercept) | 10117.8652 | 1072.2711 | 0.0000 |
| Open (habitat) | -1005.4248 | 1628.8967 | 0.5371 |
| Neandertals (type) | 1557.9881 | 1595.5904 | 0.3288 |
| Open-Neandertals | 1534.3107 | 2413.7850 | 0.5250 |
| C. | Fit | Lower | Upper |
| Neandertals (open) | 12204.739 | 9470.695 | 14938.784 |
| Neandertals (mixed) | 10893.914 | 8470.442 | 13317.385 |
| EMH (open) | 9069.408 | 6554.452 | 11584.363 |
| EMH (mixed) | 9864.755 | 7665.490 | 12064.019 |

**Table 10. Results of the two-way ANOVA (A), robust regression (B), and 95% confidence intervals (C) for mean *epLsar* given location (Central Europe, Western Europe, and Southwest Asia) and hominin type (Neandertal and early modern human).**

| A. | Estimate | Standard error | *p* value |
|---|---|---|---|
| **(Intercept)** | 3.199 e-03 | 4.785 e-04 | 4.92 e-09 |
| **Southwest Asia (location)** | -2.711 e-04 | 6.766 e-04 | 0.690 |
| **Western Europe (location)** | 2.357 e-04 | 5.998 e-04 | 0.696 |
| **Neandertals (type)** | 8.961 e-05 | 5.802 e-04 | 0.878 |
| **Southwest Asia-Neandertals** | 4.162 e-04 | 9.331 e-04 | 0.657 |
| **Western Europe-Neandertals** | -7.558 e-04 | 7.421 e-04 | 0.312 |
| **B.** | **Regression co-efficient estimate** | **Standard error** | ***p* value** |
| **(Intercept)** | 0.0029 | 0.0005 | 5.578087 e-09 |
| **Southwest Asia (location)** | 0.0001 | 0.0007 | 0.928016 |
| **Western Europe (location)** | 0.0005 | 0.0006 | 0.3946039 |
| **Neandertals (type)** | 0.0003 | 0.0006 | 0.6029024 |
| **Southwest Asia-Neandertals** | 0.0003 | 0.0010 | 0.7867205 |
| **Western Europe-Neandertals** | -0.0011 | 0.0008 | 0.1584323 |
| **C.** | **Fit** | **Lower** | **Upper** |
| **Neandertals (Central Europe)** | 0.0033 | 0.0026 | 0.0039 |
| **Neandertals (Western Europe)** | 0.0028 | 0.0022 | 0.0033 |
| **Neandertals (Southwest Asia)** | 0.0034 | 0.0023 | 0.0045 |
| **EMH (Central Europe)** | 0.0032 | 0.0022 | 0.0042 |
| **EMH (Western Europe)** | 0.0034 | 0.0027 | 0.0042 |
| **EMH (Southwest Asia)** | 0.0029 | 0.0020 | 0.0039 |

**Table 11. Results of the two-way ANOVA (A), robust regression (B), and 95% confidence intervals (C) for mean *Tfv* given location (Central Europe, Western Europe, and Southwest Asia) and hominin type (Neandertal and early modern human).**

| A. | Estimate | Standard error | *p* value |
|---|---|---|---|
| **(Intercept)** | 9278.37 | 1575.55 | 1.28 e-07 |
| **Southwest Asia (location)** | 854.39 | 2228.16 | 0.703 |
| **Western Europe (location)** | 29.77 | 1975.05 | 0.988 |
| **Neandertals (type)** | -607.10 | 1910.63 | 0.752 |
| **Southwest Asia-Neandertals** | -102.10 | 3072.89 | 0.974 |
| **Western Europe-Neandertals** | 2723.79 | 2443.70 | 0.269 |
| **B.** | **Regression co-efficient estimate** | **Standard error** | ***p* value** |
| **(Intercept)** | 9278.3732 | 1626.9853 | 1.178545 e-08 |
| **Southwest Asia (location)** | 1372.1555 | 2300.9046 | 0.5509382 |
| **Western Europe (location)** | 71.0118 | 2039.5339 | 0.9722251 |
| **Neandertals (type)** | -467.0514 | 1973.0094 | 0.8128738 |
| **Southwest Asia-Neandertals** | 404.2428 | 3173.2142 | 0.8986300 |
| **Western Europe-Neandertals** | 2643.8750 | 2532.5473 | 0.2965045 |
| **C.** | **Fit** | **Lower** | **Upper** |
| **Neandertals (Central Europe)** | 8671.276 | 6515.108 | 10827.445 |
| **Neandertals (Western Europe)** | 11424.836 | 9529.458 | 13320.215 |
| **Neandertals (Southwest Asia)** | 9423.565 | 5794.192 | 13052.938 |
| **EMH (Central Europe)** | 9278.373 | 6135.244 | 12421.503 |
| **EMH (Western Europe)** | 9308.145 | 6932.163 | 11684.127 |
| **EMH (Southwest Asia)** | 10132.765 | 6989.635 | 13275.894 |

**Table 12. Results of the Kolmogorov-Smirnov tests.**

| A. | D statistic | P value |
|---|---|---|
| **Neandertals vs. EMH for *epLsar*** | 0.13333 | 0.9062 |
| **Neandertals vs. EMH for *Tfv*** | 0.15556 | 0.7764 |
| **B.** | | |
| **Open Ntl vs. Open EMH for *epLsar*** | 0.51049 | 0.0896 |
| **Open Ntl vs. Open EMH for *Tfv*** | 0.38462 | 0.3414 |
| **C.** | | |
| **Western Europe Ntl vs. W. Europe EMH for *epLsar*** | 0.3961 | 0.1364 |
| **Western Europe Ntl vs. W. Europe EMH for *Tfv*** | 0.29221 | 0.4582 |
| **Central Europe Ntl vs. C. Europe EMH for *epLsar*** | 0.27206 | 0.8155 |
| **Central Europe Ntl vs. C. Europe EMH for *Tfv*** | 0.23529 | 0.924 |
| **Southwest Asia Ntl vs. SW Asia EMH for *epLsar*** | 0.33333 | 0.8407 |
| **Southwest Asia Ntl vs. SW Asia EMH for *Tfv*** | 0.29167 | 0.9324 |

The Tigara lived at Point Hope, Alaska from 750–250 BP, in an arid, Arctic environment that was coastal and largely without trees [138]. They relied on a diet consisting primarily of sea mammals, including whales, but supplemented with caribou, fish, birds, and edible plants [84, 130]. They are recorded ethnographically as using their anterior teeth as a third hand for processing and softening animal hides and making sinew thread [84, 128]. These tooth-use behaviors are reflected in their moderately low anisotropy and moderately high textural fill volume values [88].

The habitat conditions between the early modern humans and Point Hope Tigara would not have been tremendously different, as they both inhabited environments that were either treeless or partially forested. Although sea mammal hunting is not well documented for early modern humans, there is evidence that Upper Paleolithic humans exploited aquatic resources, such as fish, mollusks, and birds, as did the Tigara [12, 139, 140]. The Tigara required clothing for protective purposes, with animal hides and sinew serving as the raw material, and the same need for thermal protection among European Upper Paleolithic humans is probable. Indeed, there is evidence for the use of clothing for protective purposes from parallels between the mammalian taxa found in European Upper Paleolithic archaeological sites and those taxa reported in the ethnographic record as sources of fur, hide, sinew, and other raw materials that are used in the manufacture of clothing [23]. Likewise, there is ample archaeological and bio-mechanical support for the use of protective footwear [141, 142] as well as depictions of clothing and footwear, evidence of textile production, and reflections of clothing in spatial distribution of artifacts in burial contexts [141–144].

Taken together, these data suggest the early modern humans sampled here were participating in non-dietary anterior tooth-use behaviors overall and those behaviors did not differ significantly by habitat type, location, or time interval. These texture values are most similar to that of the Point Hope Tigara, a bioarchaeological sample that used their anterior dentition for clamping and grasping hides for clothing and sinew thread production. Thus, it is proposed that the early modern human and Tigara samples were participating in analogous forms of tooth-use behaviors, such as grasping and clamping hides for the production of clothing or other protective coverings.

## Early modern humans versus Neandertals

While it is possible that differences in anterior tooth-use behaviors existed between Neandertals and early modern humans, the data presented here provide no statistically significant

evidence for it (Tables 4, 5 and 7–12). Indeed, the mean anisotropy and textural fill volume values of both fossil samples reveal nearly identical results, and indicate that, as a whole, the Neandertals and early modern humans analyzed here were not engaging in vastly different tooth-use behaviors. Their anisotropy mean values are low, and within the range of non-dietary anterior tooth-use behaviors, while the textural fill volume values are fairly high, indicating a heavy bite force was required to complete these tasks. When compared to the modern human groups of known or inferred behaviors, both fossil samples align most closely with that of the Point Hope Tigara and, to a lesser extent, the Nunavut Sadlermiut (Table 4, Fig 2).

The coastal Tigara, as described above, participated in clamping and grasping tooth-use behaviors associated with hide processing and softening. However, the Nunavut Sadlermiut from northwest Hudson Bay were an interior Arctic group that relied on caribou, seal, birds, and fish [145–149]. They were inferred from archaeological remains, antemortem tooth loss, and tooth wear to have participated in extensive dental clamping and grasping behaviors for hide preparation for the production of clothing and other protective coverings [132, 145, 146]. This inferred non-dietary anterior tooth-use behavior is also supported by the microwear textures, with their extremely low anisotropy values, indicative of extensive tooth-use activities, and their very high textural fill volume values, indicating these activities required a heavy bite force.

The Point Hope Tigara sample provides the most comparable anisotropy and textural fill volume pattern to those of the two fossil samples; however, both fossil samples have higher textural fill volume values than that of the Tigara, but they are lower than that of the Sadlermiut (Table 4). Thus, a parsimonious approach is to use both bioarchaeological samples to better interpret the fossil data presented here.

Overall, the data indicate the early modern humans and Neandertals were participating in similar non-dietary anterior tooth-use activities. Using the comparative bioarchaeological datasets, those activities may be clamping and grasping behaviors for hide preparation and clothing production. These activities would have required a heavy bite force that was more than that used by the Tigara, but less than that of the Sadlermiut. As the Tigara and Sadlermiut differed in the frequency or intensity of clamping and grasping behaviors, perhaps it can be said the fossil groups were intermediate in how regularly or intensely they performed these tasks.

In what seems to be the noticeable theme of these data, there were also no significant differences in anisotropy and textural fill volume between these two hominins by habitat type nor location (Fig 2; Tables 5 and 8–12). Indeed, there is extensive overlap in values between the hominin subsamples, with variation among some of the mean values largely driven by a few outliers. For example, while the mixed-vegetation groups are nearly identical in their mean anisotropy and textural fill volume values, those for the open-vegetation are more disparate (Table 5). The possibility exists that there were some tooth-use differences between open-vegetation Neandertals and open-vegetation early modern humans (Tables 5, 8A and 12B), with the Neandertals participating in intense clamping and grasping behaviors and the early modern humans only using their anterior teeth for incising food items. However, substantial overlap is seen in their individual values, with the early modern human subsample showing more variation in values, and both subsamples having a few outliers driving the means (Fig 2).

Behavioral ingenuity between Neandertals and early modern humans can be supported or refuted depending on the dataset at hand; however, the microwear textures provide some important insight into the debate. Generally speaking, early modern humans and Neandertals sampled here participated in similar non-dietary anterior tooth-use behaviors that required a heavy bite force. Using a variety of bioarchaeological comparative samples, both the early modern humans and Neandertals closely align in texture values to those of the Tigara and

Sadlermiut, two Arctic samples that participated in clamping and grasping behaviors associated with hide preparation and processing. Continued research into this debate will inevitably lead to more robust sample sizes and strengthened interpretations; however, the datasets here support the notion that regarding non-dietary anterior teeth use Neandertals and early modern humans were not as behaviorally distinct as once considered.

## Supporting information

**S1 File. Supplementary information for each early modern human fossil used here.**
(DOCX)

**S2 File. R code for all the statistical analyses and Fig 2 plots.**
(HTML)

## Acknowledgments

We would like to thank Prof. Peter Ungar, University of Arkansas, for access to the confocal microscope, advice on the data, and continued encouragement to KLK. We acknowledge and thank the curators at the American Museum of Natural History, New York; US Museum of Natural History, Washington DC; Canadian Museum of History, Gatineau; and Natural History Museum, London, for permission to mold dental remains in their care. A particular acknowledgment goes to the Inuit Heritage Trust for permission to examine the Prince Rupert Harbour Tsimshian and Nunavut Sadlermiut dental remains. We gratefully acknowledge Sireen El Zaatari and Erik Trinkaus for their assistance in obtaining some of the fossil casts used here and to Amy Hubbard, Sarah Lacy, Maja Šešelj, and two anonymous reviewers for helpful comments on previous versions of this manuscript.

## Author Contributions

**Conceptualization:** Kristin L. Krueger.

**Data curation:** Kristin L. Krueger, Jean-Jacques Hublin, Alejandro Pérez-Pérez.

**Formal analysis:** Kristin L. Krueger, John C. Willman, Gregory J. Matthews.

**Funding acquisition:** Kristin L. Krueger.

**Investigation:** John C. Willman.

**Methodology:** Kristin L. Krueger, John C. Willman, Gregory J. Matthews.

**Resources:** Jean-Jacques Hublin, Alejandro Pérez-Pérez.

**Software:** Gregory J. Matthews.

**Supervision:** Jean-Jacques Hublin, Alejandro Pérez-Pérez.

**Visualization:** John C. Willman, Gregory J. Matthews.

**Writing – original draft:** Kristin L. Krueger, John C. Willman.

**Writing – review & editing:** Kristin L. Krueger, John C. Willman, Gregory J. Matthews, Jean-Jacques Hublin, Alejandro Pérez-Pérez.

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
