## [Decision Letter · Decision Letter 0]

23 Jul 2019

PONE-D-19-17387

Overlapping manipulative behaviors among early modern humans and Neandertals

PLOS ONE

Dear Dr. Krueger,

Thank you for submitting your manuscript to PLOS ONE. We invite you to submit a revised version of the manuscript that addresses the points raised during the review process.

Both reviewers found this paper to be interesting and innovative. They also comment on the excellent dataset for AMH and Neanderthals and the comparative modern human samples, and the sound statistical analyses that were used.

The reviewers outline certain major points, including the need to revise some of the graphs and figures, and the way in which the groups are selected.

Reviewer 2 raises specific concerns regarding the theoretical aspects and the interpretation of the results in the discussion section. The reviewer also suggests to cite some key  relevant and up to date quantitative studies.

The reviewer also requires clarification relating the claim of symbolism and the use of the anterior dentition as a tool and the correlation between vegetation and daily task activities.

The reviewer also points out to the need to highlight some of the limitation of the study, given the fact that this is an innovative approach.

We would appreciate receiving your revised manuscript by Sep 06 2019 11:59PM. To enhance the reproducibility of your results, we recommend that if applicable you deposit your laboratory protocols in protocols.io, where a protocol can be assigned its own identifier (DOI) such that it can be cited independently in the future. For instructions see: http://journals.plos.org/plosone/s/submission-guidelines#loc-laboratory-protocols

We look forward to receiving your revised manuscript.

Kind regards,

Ron Pinhasi

Academic Editor

PLOS ONE

Journal Requirements:

Reviewers' comments:

Reviewer's Responses to Questions

**Comments to the Author**

1. Is the manuscript technically sound, and do the data support the conclusions?

Reviewer #1: Yes

Reviewer #2: Partly

2. Has the statistical analysis been performed appropriately and rigorously? 

Reviewer #1: Yes

Reviewer #2: Yes

3. Have the authors made all data underlying the findings in their manuscript fully available?

Reviewer #1: Yes

Reviewer #2: Yes

4. Is the manuscript presented in an intelligible fashion and written in standard English?

Reviewer #1: Yes

Reviewer #2: Yes

5. Review Comments to the Author

Reviewer #1: This is a bold paper that seeks to compare early modern humans, Neandertals and seven bioarchaeology groups using two measures of enamel surface texture rendered from dental microwear texture analysis. The use of the incisors to unlock the behavioral repertoire of fossil hominins is provocative and the authors use an innovative approach—texture analysis coupled with an elegant statistical treatment of this exceptional dataset—to reconstruct paramasticatory actions relating to food and material processing. Although behavioral superiority is not necessarily directly tested by the incisal microwear textures, the results suggest that early modern humans and Neandertals utilized their incisors in a similar manner and the closest bioarchaeological approximation is the Point Hope Tigara and the Nunavut Territory Sadlermiut. The paper is profound in its implications backed up by enviable samples, both fossil and bioarchaeological, and I fully endorse the publication of this submission after minor revisions.

Regarding the results, the data graphs are a bit challenging to decipher—consider, if possible, demoting the bioarchaeology populations to thin dark blue ellipse lines and leave the two fossil samples as they are now, thicker lines of different brighter colors. This might add clarity to the main focus of the paper—the comparison between the two fossil hominins.

Although there are no significant differences between groups, when looking at Table 4, it seems to me that for Tfv, there are three groupings. A heavily incised group includes (1) early modern humans, Neandertals and Nunavut; (2) a medium values group includes Tigara, Puye Pueblo and Prince Rupert Tsimshian; and (3) a low values group comprising Andaman Islanders, Arikara and Island Chumash. What I would say is that recent humans can be divided into three groups but the fossils are exclusively found in the high value group, the only equivalent might be the Nunavut of the Arctic, and possibly the Tigara. For epLsar, there are two groups in my view. One group would include Nunavut and PR Tsimshian with low values and the other group would include everyone else with more elevated values.

The authors might want to consider briefly acknowledging the subjectivity of the groups. For example, Western Europe (mostly France) is quite far from Central Europe but the distance is not nearly as far as it is to SW Asia, so these groups are not necessarily equivalent in terms of location. The ecogeography would also differ between continental Europe and the near East. It might just as easily be possible to combine the Western and Central Europe groupings into one and compare this group to SW Asia. The same holds true for the habitat categories. These are minor points, however, and should be taken as suggestions for improving an already excellent paper.

Specific comments:

Title: change “overlapping” to “comparison of” or remove and shorten title

Line 62: consider citing Power and Williams (2018) at the end of the sentence and introduce food processing into the sentence: “improved food storage and processing capabilities.”

70: replace “this hominin” with “Neandertal”

84: change punctuation to “them.”

94: remove “bodies—specifically”

95: remove dash

110: replace “variations” with “variation”

116: change “showed” to “show” to agree with the verb tense of the rest of this section

120: consider removing the first sentence containing “from the dialogue” for clarity

121: In the abstract, the authors use EMH, but switch to “early modern humans” for the rest of the paper—I like the latter

much better than the abbreviation—the authors may want to replace “EMH” with “early modern humans” in the abstract and elsewhere

127-8: Many early modern humans from Europe and SW Asia also exhibit shoveling

129: “average early modern human” seems vague—be more specific here

139: don’t need to mention the abbreviation “LSAMAT” since you only mention it once

147: replace “expound” with “identify”

151: remove “focal” to avoid confusion with focal animal sampling or ethnographic usages (e.g., Hewlett, 1991, Intimate Fathers)

Table 2: consider replacing “time” with “MIS” as it will increase transparency

166: “own unique” seems redundant

166-172: consider removing this paragraph, except the last half of the last sentence (see below)

172-174: Start next paragraph with “The fossil sample was examined by habitat, location and time interval.”—this would be the first sentence

192: change punctuation for “open,” “mixed,”

178-190: perhaps provide a stronger justification for the category of “mixed” as a wide range of habitats are considered

203-204: improve justification for the lumping of MIS 7-5 and MIS 4-3 into distinct groups, such as ecology or climate extremes

351: change “possible” to “possibly”

352: remove sentence beginning with “Each analysis will be…”

366: Remove “examining it through the lens of”

416: Change “was” to “were”

417: insert “dental” between “extensive” and “clamping”

426: replace “conservative” with “parsimonious”

428: replace “indicates” with “indicate” because it’s modifying “data” (plural)

436: consider removing the paragraph beginning on 436 to avoid redundancy

447: Combine “Discussion and Conclusions” into a single section and remove conclusion paragraph beginning on 447 to

avoid redundancy

Literature

Power RC, Williams FL (2018) The increasing intensity of food processing during the Upper Paleolithic of western Eurasia. Journal of Paleolithic Archaeology 1:281–301. http://dx.doi.org/10.1007/s41982-018-0014-x.

Reviewer #2: The authors analysed the anterior tooth wear in Early Modern Humans (EMH) and Neanderthals using dental microwear texture analysis. They found no statistically significant differences between two hominin groups, suggesting a similar non-dietary use of their anterior teeth. The manuscript is overall well-written, and based on a good sample size including a large comparative modern human group. While the study is certainly interesting, I feel the authors did not fully exploit their results. This study generally lacks a detailed discussion on several important aspects related to anterior tooth wear. For instance, while the study focused on the “anterior dental loading hypothesis” revolving around the Neanderthal dentition, the authors failed to acknowledge this important aspect. They never mentioned this hypothesis throughout the entire manuscript.

Moreover, the problem with this particular study is that it does not present anything original or unexpected. Similar results have been already presented in early works. Therefore, the authors need to highlight more what is new in their study. The authors should probably create another section in the Discussion, where they discuss, biomechanically, what a heavy anterior tooth wear can tell us. For example, the authors acknowledge that textural fill volume values are fairly high, indicating a heavy bite force. However, they never truly discuss this fundamental aspect of their result. This difference could also be related to variation in enamel thickness between the two human species. This aspect should be also considered.

There is also a general lack of key references throughout the manuscript (see a list of importance missing references below). There are many old studies, often in a different language, that are cited throughout the manuscript. These studies probably add very little information on how Neanderthals and AMH were using their anterior teeth. They were mostly qualitative works, and therefore they rarely accurately describe and quantify anterior dental wear in Pleistocene humans. At the same time, many critical (and more relevant to this study) and highly cited studies, were completely ignored.

Neanderthal flexible diet:

Fiorenza, L., Benazzi, S., Tausch, J., Kullmer, O., Bromage, T.G., Schrenk, F., 2011. Molar macrowear reveals Neanderthal eco-geographical dietary variation. PLoS ONE 6, e14769.

Fiorenza, L., Benazzi, S., Henry, A., Salazar-García, D.C., Blasco, R., Picin, A., Wroe, S., Kullmer, O., 2015a. To meat or not to meat? New perspectives on Neanderthal ecology. Yearbook of Physical Anthropology 156, S59, 43-71.

Non-masticatory use of teeth in Neanderthals and EMH

Fiorenza, L., Kullmer, O., 2013. Dental wear and cultural behaviour in Middle Paleolithic humans from the Near East. American Journal of Physical Anthropology 152, 107-117.

Volpato, V., Macchiarelli, R., Guatelli-Steinberg, D., Fiore, I., Bondioli, L., Frayer, D.W., 2012. Hand to mouth in a Neandertal: Right handedness in Regourdou 1. PLoS ONE 7, e43949.

Bruner E., Lozano, M.R., 2014. Extended mind and visuo-spatial integration: three hands for the Neandertal lineage. Journal of Anthropological Sciences 92, 273-280.

Biomechanical interpretation of Neanderthal anterior tooth wear

O’Connor C.F., Franciscus R.G., Holton N.E., 2005. Bite force production capability and efficieny in Neandertals and modern humans. Am. J. Phys. Anthropl. 127, 129-151

Anton S.C., 1990. Neandertals and the anterior dental loading hypothesis: a biomechanical evaluation of bite force production. Kroeber Anthropol Soc Pap 71-72, 67-76.

Anton S.C., 1994. Mechanical and other perspectives on Neanderthal craniofacial morphology. In” Corruccini R.S., Ciochion R.L., editors. Integrative paths to the past. Englewood Cliffs: Prenctice Hall, pp. 677-795.

Wroe S., Parr W.C.H., Ledogar J.A., Bourke J., Evans, S.P., Fiorenza L., Benazzi S., Hublin J-J., Kullmer O. and T. Yokley, 2018. Computational simulations show that Neanderthal facial morphology represents adaptation to cold and high energy demands, but not heavy biting. Proc. R. Soc. Lond. B Biol. Sci., DOI: 10.1098/rspb.2018.0085

I generally do not understand the connection between anterior tooth wear and symbolic behaviour. Is there anything symbolic in using your teeth as tools?

I also do not understand the correlation between vegetation and daily task activities. This should be further expanded and discussed.

The authors never considered in their study the division of labor in daily task activities between males and females. For instance, Estalrrich et al. (2015) found tooth wear differences between males and females in the Neanderthals from three different sites.

Finally, authors should further highlight the limitations of their study, in terms of sample size and methodology. For example, microwear can change very rapidly, within weeks, or even days. Therefore, the interpretation of the microwear signal could be wrongly interpreted.

6. PLOS authors have the option to publish the peer review history of their article (what does this mean?). If published, this will include your full peer review and any attached files.

Reviewer #1: No

Reviewer #2: No

---

## [Author Response · Author response to Decision Letter 0]

20 Aug 2019

Reviewer #1: This is a bold paper that seeks to compare early modern humans, Neandertals and seven bioarchaeology groups using two measures of enamel surface texture rendered from dental microwear texture analysis. The use of the incisors to unlock the behavioral repertoire of fossil hominins is provocative and the authors use an innovative approach—texture analysis coupled with an elegant statistical treatment of this exceptional dataset—to reconstruct paramasticatory actions relating to food and material processing. Although behavioral superiority is not necessarily directly tested by the incisal microwear textures, the results suggest that early modern humans and Neandertals utilized their incisors in a similar manner and the closest bioarchaeological approximation is the Point Hope Tigara and the Nunavut Territory Sadlermiut. 

Note: we changed “behavioral superiority” to “behavioral ingenuity”. 

The paper is profound in its implications backed up by enviable samples, both fossil and bioarchaeological, and I fully endorse the publication of this submission after minor revisions.

Regarding the results, the data graphs are a bit challenging to decipher—consider, if possible, demoting the bioarchaeology populations to thin dark blue ellipse lines and leave the two fossil samples as they are now, thicker lines of different brighter colors. This might add clarity to the main focus of the paper—the comparison between the two fossil hominins.

We did not use bioarchaeological samples in any of the graphs, but did include break-downs of the fossil samples (e.g. by habitat, location, and time). However, we agree with Reviewer 1 that these breakdown graphs were confusing to read, and have removed them. We kept the one data graph that shows the early modern human and Neandertals samples only, and then added in new data graphs showing each individual bioarchaeological comparative group with the two fossil groups (i.e. Neandertals and early modern humans). We decided to do this after adding in all the comparative groups in one graph, and it suffered from the same problems as the others – too many groups in one graph – so we split them up. 

Although there are no significant differences between groups, when looking at Table 4, it seems to me that for Tfv, there are three groupings. A heavily incised group includes (1) early modern humans, Neandertals and Nunavut; (2) a medium values group includes Tigara, Puye Pueblo and Prince Rupert Tsimshian; and (3) a low values group comprising Andaman Islanders, Arikara and Island Chumash. What I would say is that recent humans can be divided into three groups but the fossils are exclusively found in the high value group, the only equivalent might be the Nunavut of the Arctic, and possibly the Tigara. For epLsar, there are two groups in my view. One group would include Nunavut and PR Tsimshian with low values and the other group would include everyone else with more elevated values.

We appreciate the reviewer’s attention to these data. KLK consulted with the statistician on the paper, (Greg Matthews), and he suggested doing density plots 1. because we are only looking at one variable at a time (Tfv or epLsar) and 2. to see how the data fall. These density plots (see below) use a kernel density estimate to show probability, and are a smoothed histogram. Tfv only shows two groups (high and low), while epLsar peaks and perhaps shows other groupings, but it’s not clear or obvious. As a result, we did not continue with further analysis. 

The authors might want to consider briefly acknowledging the subjectivity of the groups. For example, Western Europe (mostly France) is quite far from Central Europe but the distance is not nearly as far as it is to SW Asia, so these groups are not necessarily equivalent in terms of location. The ecogeography would also differ between continental Europe and the near East. It might just as easily be possible to combine the Western and Central Europe groupings into one and compare this group to SW Asia. The same holds true for the habitat categories. These are minor points, however, and should be taken as suggestions for improving an already excellent paper.

Our finer breakdown by habitat and chronology in addition to location adds a level of variation that is not addressed simply by looking at Western/Central Europe vs SW Asia. The graphic displays by location are also quite clear, in that, one can look at either Western or Central Europe vs SW Asia without the need for creating the European macrogroup. We appreciate this suggestion but think our choice of subgroups, following our previously published methodology (Krueger et al., 2017), sufficiently covers variation within our sample. That said, we respect the reviewer’s concern, and have added in a paragraph in the Materials & Methods section that discusses the broad nature of the groupings, the possible limitations of them, and additional references were added (Fiorenza et al., 2011; Williams et al., 2018) to emphasize that other studies utilize different grouping methods. 

Specific comments:

Title: change “overlapping” to “comparison of” or remove and shorten title 

We modified the title for clarity.

Line 62: consider citing Power and Williams (2018) at the end of the sentence and introduce food processing into the sentence: “improved food storage and processing capabilities.”

Citation added – thank you for bringing this citation to our attention.

70: replace “this hominin” with “Neandertal” Done.

84: change punctuation to “them.” Done

94: remove “bodies—specifically” Done

95: remove dash Done

110: replace “variations” with “variation” Done

116: change “showed” to “show” to agree with the verb tense of the rest of this section Done

120: consider removing the first sentence containing “from the dialogue” for clarity Sentence re-written for clarity.

121: In the abstract, the authors use EMH, but switch to “early modern humans” for the rest of the paper—I like the latter

much better than the abbreviation—the authors may want to replace “EMH” with “early modern humans” in the abstract and elsewhere 

While we generally agree with this statement, sometimes using “early modern human” in a sentence gets too cumbersome and wordy. We have left it as EMH, but understand from where the reviewer is coming.

127-8: Many early modern humans from Europe and SW Asia also exhibit shoveling 

We agree with this comment. We intentionally used the word “frequently” to indicate that overlap exists with early modern humans. However, the frequency of mass-additive characters is much greater in archaic humans than in Middle Paleolithic modern humans or Upper Paleolithic modern humans. Another citation has been added which includes tables showing these frequency differences using recently tabulated data from the literature and the Sima de las Palomas Neandertals (Zapata et al., 2017). Additional traits have also been listed in the manuscript text. The additional citation should clarify the meaning of frequency differences better than the previous citation alone. 

Zapata, J., Bayle, P., Lombardi, A. V., Pérez-Pérez, A., & Trinkaus, E. (2017). The Palomas dental remains: preservation, wear and morphology In E. Trinkaus, & M. J. Walker (Eds.), The People of Palomas: Neandertals from the Sima de las Palomas, Cabezo Gordo, Southeastern Spain (pp. 52-104). College Station: Texas A&M University Press.

129: “average early modern human” seems vague—be more specific here 

We have rephrased this in the text and added citations. 

139: don’t need to mention the abbreviation “LSAMAT” since you only mention it once Abbreviation removed

147: replace “expound” with “identify” Done

151: remove “focal” to avoid confusion with focal animal sampling or ethnographic usages (e.g., Hewlett, 1991, Intimate Fathers) Done

Table 2: consider replacing “time” with “MIS” as it will increase transparency 

We worry that changing “time” with “MIS” in this table in particular will be confusing to the reader since our categories are based on MIS designations, but don’t use them specifically. We have, instead, changed this column name from “Time” to “Chronology”.

166: “own unique” seems redundant Agreed. We removed “unique”.

166-172: consider removing this paragraph, except the last half of the last sentence (see below) Done

172-174: Start next paragraph with “The fossil sample was examined by habitat, location and time interval.”—this would be the first sentence Done

192: change punctuation for “open,” “mixed,” Done

178-190: perhaps provide a stronger justification for the category of “mixed” as a wide range of habitats are considered 

We understand the reviewer’s concerns, and have added in a paragraph that discusses the subjectivity of the groupings. We invite researchers to evaluate these datasets, and have provided the raw data for each fossil in the S1 file and SOM in Krueger et al., 2017.

203-204: improve justification for the lumping of MIS 7-5 and MIS 4-3 into distinct groups, such as ecology or climate extremes The fossil samples were not compared by time since the groupings were different. We have clarified this in the manuscript.

351: change “possible” to “possibly” Done

352: remove sentence beginning with “Each analysis will be…” Done

366: Remove “examining it through the lens of” Done

416: Change “was” to “were” Done

417: insert “dental” between “extensive” and “clamping” Done

426: replace “conservative” with “parsimonious” Done

428: replace “indicates” with “indicate” because it’s modifying “data” (plural) Done

436: consider removing the paragraph beginning on 436 to avoid redundancy See next comment

447: Combine “Discussion and Conclusions” into a single section and remove conclusion paragraph beginning on 447 to

avoid redundancy 

We changed the heading to “Discussion and Conclusions” and removed the last paragraph of the discussion section to avoid redundancy.

Literature

Power RC, Williams FL (2018) The increasing intensity of food processing during the Upper Paleolithic of western Eurasia. Journal of Paleolithic Archaeology 1:281–301. http://dx.doi.org/10.1007/s41982-018-0014-x. Reference added.

Reviewer #2: The authors analysed the anterior tooth wear in Early Modern Humans (EMH) and Neanderthals using dental microwear texture analysis. They found no statistically significant differences between two hominin groups, suggesting a similar non-dietary use of their anterior teeth. The manuscript is overall well-written, and based on a good sample size including a large comparative modern human group. While the study is certainly interesting, I feel the authors did not fully exploit their results. This study generally lacks a detailed discussion on several important aspects related to anterior tooth wear. 

For instance, while the study focused on the “anterior dental loading hypothesis” revolving around the Neanderthal dentition, the authors failed to acknowledge this important aspect. They never mentioned this hypothesis throughout the entire manuscript. 

We want to respectfully address the reviewer’s concerns about our lack of discussion surrounding biomechanics of Neandertal anterior tooth wear. We acknowledge that there are multiple ways to address anterior tooth-use behaviors in the fossil record, including exploring aspects of functional adaptation and morphological evolution, and bioarchaeological/comparative approaches for behavioral reconstructions. We have taken the latter approach to frame our paper. 

This bioarchaeological/comparative approach to anterior tooth-use has been published elsewhere for bioarchaeological groups and Late Pleistocene humans (El Zaatari et al., 2014; Hlusko et al., 2013; Krueger, 2014, 2015, 2016; Krueger and Ungar, 2009, 2012; Krueger et al., 2017) without a need to include discussions surrounding the Anterior Dental Loading Hypothesis (ADLH). The statement that we “failed to acknowledge” infers that we intended to write a paper concerned with biomechanical implications, which we did not. 

We agree that there are (potential) biomechanical and/or morphological implications of our behavioral reconstructions, and readers are welcome to use our datasets and analysis for these types of interpretations, if they so choose. However, biomechanical analyses are often at odds with each other, including the literature surrounding Neandertal and early modern human craniodental morphology. Indeed, there is a clear debate in the literature regarding this topic, with some supporting and others refuting the ADLH. This hypothesis is based on an assumption that observations of craniodental morphology and tooth-use (inferred from wear and morphology) are interrelated, asserting that craniodental morphology is functionally adapted to high and/or repetitive anterior dental loading.

Inferences drawn from our results concerning morphological evolution (and biomechanical consequences) are inherently speculative. We seek to avoid such speculation, focus on the dataset at hand, and interpret our results using a strong comparative framework. 

Krueger KL, & Ungar PS. (2009). Incisor microwear textures of five bioarcheological groups. International Journal of Osteoarchaeology, 20(5), 549-560.

Krueger KL, & Ungar PS. (2012). Anterior dental microwear texture analysis of the Krapina Neandertals. Central European Journal of Geosciences, 4(4), 651-662.

Hlusko LJ, Carlson JP, Guatelli-Steinberg D, Krueger KL, Mersey B, Ungar PS, et al. (2013). Neanderthal teeth from Moula-Guercy, Ardèche, France. American Journal of Physical Anthropology, 151(3), 477-491, doi:10.1002/ajpa.22291.

El Zaatari S, Krueger KL, & Hublin J-J. (2014). Dental microwear texture analysis and the diet of the Scladina I-4A Neandertal child. In M. Toussaint, & D. Bonjean (Eds.), The Scladina I-4A Juvenile Neandertal (Andenne, Belgium): Palaeoanthropology and Context (pp. 363-378). Liège: Études et Recherches Archéologiques de Université de Liège.

Krueger KL. (2014). Contrasting the Ipiutak and Tigara: Evidence from incisor microwear texture analysis. In C. E. Hilton, B. M. Auerbach, & L. W. Cowgill (Eds.), The Foragers of Point Hope: The Biology and Archaeology of Humans on the Edge of the Alaskan Arctic (pp. 99-119). Cambridge: Cambridge University Press.

Krueger KL. (2015). Reconstructing diet and behavior in bioarchaeological groups using incisor microwear texture analysis. Journal of Archaeological Science: Reports, 1, 29-37.

Krueger KL. (2016). Dentition, behavior, and diet determination. In JD Irish, & GR Scott (Eds.), A Companion to Dental Anthropology (pp. 396-411). Malden: John Wiley & Sons, Inc.

Krueger KL, Ungar PS, Guatelli-Steinberg D, Hublin J-J, Pérez-Pérez A, Trinkaus E, et al. (2017). Anterior dental microwear textures show habitat-driven variability in Neandertal behavior. Journal of Human Evolution, 105, 13-23, doi: http://dx.doi.org/10.1016/j.jhevol.2017.01.004.

Moreover, the problem with this particular study is that it does not present anything original or unexpected. Similar results have been already presented in early works. 

We disagree with this assessment of our study. There has never been an analysis comparing Neandertal and early modern human anterior tooth-use with dental microwear texture analysis. Dental microwear texture analysis provides the most objective method for characterizing and quantifying dental microwear, allowing for an improved resolution in our analysis of Neandertal and early modern human tooth-use behaviors. Additionally, using comparative biorchaeological samples of known or inferred tooth-use behaviors to estimate the actual behaviors employed by these fossil individuals has not been completed before. Considering the cultural stereotype of Neandertals being a “less than” hominin, this study shows another line of evidence to the contrary. 

Therefore, the authors need to highlight more what is new in their study. The authors should probably create another section in the Discussion, where they discuss, biomechanically, what a heavy anterior tooth wear can tell us. For example, the authors acknowledge that textural fill volume values are fairly high, indicating a heavy bite force. However, they never truly discuss this fundamental aspect of their result. This difference could also be related to variation in enamel thickness between the two human species. This aspect should be also considered.

We disagree with the need for a discussion on biomechanics given the reasoning above. While a biomechanical approach is one avenue to take, we chose to take a comparative approach. 

Re: heavy bite force – This idea comes from previous assessments of microwear and cranial biomechanics (e.g., Ungar and Spencer, 1999; Spencer and Ungar 2000). However, this “heavy bite force” is inferred through a comparative framework of bioarchaeological groups in the present study without concomitant analyses of cranial biomechanics in the same groups. Given that multiple groups with highly variable morphologies exhibit “heavy bite force” microwear texture signatures, we can infer that this is a behavioral signature of habitual heavy loading rather than a strict function of how much load could be produced based on the functional morphology of an individual. See examples in Corruccini (1999) for examples of idiosyncratic variation in bite force within a contemporary human population. 

Re: enamel thickness - We agree that there are hypotheses concerning the relationship between anterior dental enamel thickness (or tissue proportions) and how they would react to/dissipate biomechanical forces produced. While some researchers choose to create a model for the production/dissipation of bite force relative to enamel thickness (or tissue proportions), others record the actual signatures of loading behavior through dental microwear textures. We have taken the latter approach.

We took this approach for several reasons. There exists no comparative data on dental tissues for our bioarchaeological human groups and limited data for the Pleistocene fossils used here. Additionally, many of the teeth examined here exhibit some degree of occlusal wear. Wear changes the biomechanical properties of a tooth (in terms of bite force dissipation), and influences occlusal relationships throughout the lifespan. As a result, wear and occlusion are confounding factors for biomechanical analyses. For these reasons, we would not be creating a rigorous model of the relationship between enamel/dentin/pulp tissue proportions and bite force production in Late Pleistocene and recent humans from our dental microwear texture data, leaving any inferences as speculative at best. 

We hope the behavioral data presented here will provide a springboard for others to study the potential co-variation between these confounding factors; however, this topic is outside the scope of this manuscript. 

Corruccini RS. (1999). How anthropology informs the orthodontic diagnosis of malocclusion's causes: Edwin Mellen Press.

Ungar PS, & Spencer MA. (1999). Incisor microwear, diet, and tooth use in three Amerindian populations. American Journal of Physical Anthropology, 109(3), 387-396.

Spencer MA, & Ungar PS. (2000). Craniofacial morphology, diet and incisor use in three native American populations. International Journal of Osteoarchaeology, 10(4), 229-241.

There is also a general lack of key references throughout the manuscript (see a list of importance missing references below). There are many old studies, often in a different language, that are cited throughout the manuscript. 

We acknowledge that using research published in other languages is important and contributes to our understanding of fossil hominins, especially those from outside English-speaking countries. We find them helpful to our present research to help put the fossils into original context. 

These studies probably add very little information on how Neanderthals and AMH were using their anterior teeth. They were mostly qualitative works, and therefore they rarely accurately describe and quantify anterior dental wear in Pleistocene humans. 

We agree that some past research is qualitative in nature, which is why we are taking our nuanced, quantitative approach to this phenomenon. However, it is inaccurate to suggest that the non-English, qualitative, or “old studies” are irrelevant to the present paper (or any contemporary research). For example, the anterior dental loading hypothesis (ADLH) was initially proposed, and quantitatively tested, as a result of accumulating evidence from many non-English, “old”, and/or qualitative studies. We stand by the use of these sources and feel it is important to look back at the history of our discipline as much as it is to look forward. 

At the same time, many critical (and more relevant to this study) and highly cited studies, were completely ignored. 

Neanderthal flexible diet: 

We agree that these references on Neandertal flexible diet should have been included, and have added them in discussions surrounding Neandertal diet in the “Tooth-use behaviors in the Paleolithic” section.

Fiorenza, L., Benazzi, S., Tausch, J., Kullmer, O., Bromage, T.G., Schrenk, F., 2011. Molar macrowear reveals Neanderthal eco-geographical dietary variation. PLoS ONE 6, e14769.

Fiorenza, L., Benazzi, S., Henry, A., Salazar-García, D.C., Blasco, R., Picin, A., Wroe, S., Kullmer, O., 2015a. To meat or not to meat? New perspectives on Neanderthal ecology. Yearbook of Physical Anthropology 156, S59, 43-71.

We thank the reviewer for their suggested references to include in our paper, and appreciate the time it took to provide that information. That said, we want to address the reviewer’s comment that these studies “were completely ignored.” We want to assure the reviewer that any missed references were not done intentionally. Sometimes citations are missed, and we are grateful that the reviewer brought these missed citations to our attention. 

Non-masticatory use of teeth in Neanderthals and EMH 

Fiorenza, L., Kullmer, O., 2013. Dental wear and cultural behaviour in Middle Paleolithic humans from the Near East. American Journal of Physical Anthropology 152, 107-117.

We have included this reference in our Introduction section.

Volpato, V., Macchiarelli, R., Guatelli-Steinberg, D., Fiore, I., Bondioli, L., Frayer, D.W., 2012. Hand to mouth in a Neandertal: Right handedness in Regourdou 1. PLoS ONE 7, e43949.

We have included this reference in our “Tooth use behaviors in the Paleolithic” section.

Bruner, E., & Lozano, M. (2014). Extended mind and visuo-spatial integration: three hands for the Neandertal lineage. Journal of Anthropological Sciences, 92, 273-280.

As this is tangentially related to our topic, and does not discuss specific details of anterior tooth-use behaviors (or specific examples, only provides references to specific examples), we decided not to include this reference. However, it should be noted that we do include Lozano et al., 2017, which includes the “extended mind and visuo-spatial integration hypothesis” and handedness, making the inclusion of this citation unnecessary. 

Biomechanical interpretation of Neanderthal anterior tooth wear 

O’Connor C.F., Franciscus R.G., Holton N.E., 2005. Bite force production capability and efficieny in Neandertals and modern humans. Am. J. Phys. Anthropl. 127, 129-151

Anton S.C., 1990. Neandertals and the anterior dental loading hypothesis: a biomechanical evaluation of bite force production. Kroeber Anthropol Soc Pap 71-72, 67-76.

Anton S.C., 1994. Mechanical and other perspectives on Neanderthal craniofacial morphology. In” Corruccini R.S., Ciochion R.L., editors. Integrative paths to the past. Englewood Cliffs: Prenctice Hall, pp. 677-795.

Wroe S., Parr W.C.H., Ledogar J.A., Bourke J., Evans, S.P., Fiorenza L., Benazzi S., Hublin J-J., Kullmer O. and T. Yokley, 2018. Computational simulations show that Neanderthal facial morphology represents adaptation to cold and high energy demands, but not heavy biting. Proc. R. Soc. Lond. B Biol. Sci., DOI: 10.1098/rspb.2018.0085

The selected papers proposed for inclusion promote only one side of the debate, and fail to adequately acknowledge the back-and-forth arguments on and limitations of the issue. For this reason, we will maintain neutrality on the subject, and allow this paper to be cited as others see fit. Instead, we present our direct observations of dental microwear using an analogical framework built of robust samples to allow the data to speak for themselves instead of incorporating them into existing theoretical frameworks (see also Daegling et al., 2013).

Daegling DJ, Judex S, Ozcivici E, Ravosa MJ, Taylor AB, & Grine FE, et al. (2013). Viewpoints: Feeding mechanics, diet, and dietary adaptations in early hominins. Am J Phys Anthropol. 151(3): 356-371. 

I generally do not understand the connection between anterior tooth wear and symbolic behaviour. Is there anything symbolic in using your teeth as tools?

We agree there is no connection between anterior tooth wear and symbolic behavior. The only place where symbolic behavior is mentioned is in the introduction when we discuss that Neandertals were capable of such behaviors, making them similar to early modern humans in this regard (or, not as different from early modern humans). This is making a case for Neandertal behavior being more sophisticated than originally surmised. We apologize for any confusion here. 

I also do not understand the correlation between vegetation and daily task activities. This should be further expanded and discussed. 

Vegetation coverage provides an ecological proxy. The assumption is that patterns of daily task activities vary by local needs, and local needs are a reflection the ecological setting in which an individual lives. This argument is similar to that made by Ofer Bar-Yosef (2004) in which an individual will “eat what is there.” This concept also forms the basis of most of the dietary papers that the reviewer is suggesting we add to the manuscript. In sum, behavior (dietary, manipulative, or otherwise) is likely to vary by local ecology. We recommend Binford (2001) for more information on the topic. 

Bar-Yosef O. (2004). Eat what is there: hunting and gathering in the world of Neanderthals and their neighbours. Int J Osteoarchaeol. 14(3-4), 333-342.

Binford LR. (2001). Constructing Frames of Reference: An Analytical Method for Archaeological Theory. Berkeley: University of California Press.

The authors never considered in their study the division of labor in daily task activities between males and females. For instance, Estalrrich et al. (2015) found tooth wear differences between males and females in the Neanderthals from three different sites.

We agree that this would be an important analysis, and we did consider it. While some individuals are of known sex, most are not due to the many isolated dental remains in the sample. We could not accumulate a sample size with known-sex individuals that would provide a robust statistical analysis. All stratigraphic and excavation information about each fossil used here can be found in the S1 file. Anyone that would like to pursue this topic further has access to our dataset. 

Finally, authors should further highlight the limitations of their study, in terms of sample size and methodology. For example, microwear can change very rapidly, within weeks, or even days. Therefore, the interpretation of the microwear signal could be wrongly interpreted.

We acknowledge that there are always limitations in any study, and we have provided key information about the subjectivity of our fossil groupings, as other studies have grouped their fossils differently. However, we have two concerns about Reviewer 2’s comment. The first is that the limitations of our methodology, dental microwear, have been addressed at length in several important review papers (see Ungar et al., 2008 and Krueger, 2016). Discussing the limitations of dental microwear is most appropriate for a review paper, and outside the scope of our research paper. Specifically, the reviewer raised concerns about microwear turnover, which we assume is the “Last Supper Effect” - often discussed as a limitation of occlusal molar microwear (but see Walker and Teaford, 1989). We agree that there surely is some degree of the “Last Supper Effect” on the labial surfaces of anterior teeth; however, this effect has not been tested on these tooth types. Some evidence from in vitro buccal microwear on postcanine teeth suggests that the turnover rate is slower than on the occlusal surface (e.g. Romero et al., 2012), but this is all that exists on the subject to date.

The second concern we have is the reviewer’s comment about sample size. This is an inherent limitation for any paleoanthropological analysis. It is puzzling that Reviewer 2 referred to our fossil and comparative samples as “a good sample size,” but then considered it a limitation later on in their review. Moreover, if it is a limitation, then we are introduced with the following paradox: how can the studies noted by Reviewer 2 (e.g. Fiorenza et al., 2011; Fiorenza & Kuller, 2013) use far fewer fossil individuals in their analyses, multiple teeth from the same individuals, and fewer and smaller bioarchaeological comparative groups than our study without a similar reflection on sample size? Likewise, other papers that were suggested for citation are analyses of single individuals (Volpato et al., 2012) or use only one ethnographic outgroup (e.g., Bruner and Lozano, 2014). Thus, we do not view sample size as an oversight, but an issue that is widely understood by the readership of paleoanthropological research. We respectfully disagree that an in-depth scrutiny of our fossil and comparative sample sizes is necessary. 

This is a succinct and transparent study. It is made all the more transparent by the inclusion of raw data and R code. Any researcher is welcome to reanalyze our data and add their nuanced ideas and approaches. We welcome such analyses as they help drive the discipline forward. 

Walker A & Teaford MF. (1989). Inferences from quantitative analysis of dental microwear. Folia Primatol (Basal). 53(1-4): 177-189.

Ungar PS, Scott RS, Scott JR & Teaford MF. (2008). Dental microwear analysis: historical perspectives and new approaches. In: Irish JD & Nelson GC, eds. Technique and Application in Dental Anthropology, Cambridge University Press, New York, pp. 389-425.

Krueger KL. (2016). Dentition, behavior, and diet determination. In: Irish, JD & Scott GR, eds. A Companion to Dental Anthropology, Wiley Blackwell, Malden, pp. 396-411.

Romero A, Galbany J, De Juan J, & Pérez‐Pérez A. (2012). Short-and long-term in vivo human buccal–dental microwear turnover. Am J Phys Anthropol. 148(3), 467-472.

---

## [Decision Letter · Decision Letter 1]

18 Sep 2019

PONE-D-19-17387R1

Anterior tooth-use behaviors among early modern humans and Neandertals

PLOS ONE

Dear Dr Kreuger,

Thank you for submitting your manuscript to PLOS ONE. As you can see from the included comments by the reviewers, both of them feel that the revised version did not address all the aspects raised and in addition reviewer 1 raises some additional minor points that should  be revised. . Therefore, we invite you to submit a revised version of the manuscript that addresses the points raised during the review process.

In the case of reviewer 2, I would appreciate if you can address  the comments by both making some changes to the text and in cases in which you strongly disagree, please provide a response which refutes these claims together with the  revised manuscript.

We would appreciate receiving your revised manuscript by  October 18.

To enhance the reproducibility of your results, we recommend that if applicable you deposit your laboratory protocols in protocols.io, where a protocol can be assigned its own identifier (DOI) such that it can be cited independently in the future. For instructions see: http://journals.plos.org/plosone/s/submission-guidelines#loc-laboratory-protocols

We look forward to receiving your revised manuscript.

Kind regards,

Ron Pinhasi

Academic Editor

PLOS ONE

Reviewers' comments:

Reviewer's Responses to Questions

**Comments to the Author**

1. If the authors have adequately addressed your comments raised in a previous round of review and you feel that this manuscript is now acceptable for publication, you may indicate that here to bypass the “Comments to the Author” section, enter your conflict of interest statement in the “Confidential to Editor” section, and submit your "Accept" recommendation.

Reviewer #1: All comments have been addressed

Reviewer #2: (No Response)

2. Is the manuscript technically sound, and do the data support the conclusions?

Reviewer #1: Yes

Reviewer #2: Partly

3. Has the statistical analysis been performed appropriately and rigorously? 

Reviewer #1: Yes

Reviewer #2: Yes

4. Have the authors made all data underlying the findings in their manuscript fully available?

Reviewer #1: Yes

Reviewer #2: Yes

5. Is the manuscript presented in an intelligible fashion and written in standard English?

Reviewer #1: Yes

Reviewer #2: Yes

6. Review Comments to the Author

Reviewer #1: Review of Krueger et al. for PONE:

The authors have successfully responded to the critiques and suggestions of the reviewers. The result is an improved manuscript that is nearly ready for publication. The comments below are meant to further improve this excellent analysis and comparison of Middle and Upper Paleolithic anterior tooth texture. Although the relationship between anterior tooth use and cultural inferiority / superiority is not immediately obvious, in the Discussion the authors explain better how perceptions of Neandertals can influence the explanation of the variation in tooth wear and other studies. The lack of differences imply that similar behaviors were performed by Middle and Upper Paleolithic peoples and some of these were mimicked by their Holocene counterparts. The paper will have a high impact on the field and I encourage the rapid publication of the manuscript.

Specific comments:

Introduction

• “had a more complex division of labor for resource acquisition” doesn’t quite make sense. More complex than whom? Maybe omit “more”

• “show distinctions between Neandertals and early modern humans – albeit, with some overlap” —consider revising, or perhaps replace with statistically meaningful terms such as “show a pattern in the distinctions in the mean values for Neandertal and early modern humans, but with large standard deviations resulting in nonsignificant differences…”

M&M

• Table 2: Just double checking: you investigated La Quina 1, not La Quina 5, correct? In any event, it might be interesting to eventually examine the incisal microtexture of La Quina 5.

• “under the auspices of three attainable factors:” is unclear—consider changing to “using three factors”

• Change “environmental reconstructions, dating techniques, etc” to “environmental reconstructions and dating techniques”

• “[S1 file and SOM in 37; however, these limitations resulted in broad categories. We recognize that other researchers may use different groupings [101, 102].” Is problematic. The first clause is not a complete sentence yet includes a period and the end misses a double bracket. I recommend changing it to “as shown in the S1 file [and SOM in 37]; however, these limitations resulted in broad categories [cf. 101, 102].”

• “Underpinnings” is unclear: Change “their mathematical underpinnings are described in Scott et al.” to “their mathematical descriptions are detailed in Scott et al.”

• To increase clarity, maybe replace “time), six combinations in total, a” with “time)--six combinations in total—a”

Results

• Just a recommendation—the authors might want to comb through the paper carefully and omit redundancy. For example, in the Results, I felt like l read that “Closed” habitat is removed in this analysis as no early modern humans analyzed here lived in “closed” environments.” or something like it multiple times in the text and captions—maybe remove one of them? There are a few others that are repeated once too often –maybe remove from table and figure captions?

Discussion and conclusion

• I’d recommend removing entirely “While comparisons with the Neandertal sample will be addressed in more detail below, the similarity with the Point Hope Tigara commands consideration.”

• Consider changing “in European Upper Paleolithic archaeological sites and those taxa used ethnographically for the fur, hide, sinew, and other raw materials” to “in European Upper Paleolithic archaeological sites and those taxa reported in the ethnographic record as sources of the fur, hide, sinew and other raw materials”

• “Sadlermiuuit (Table 4).: still retains track-changes and later on lines 429, 430

• Lines 439-444--I think what we are seeing here in terms of the similarity of textures is the eat what you can find phenomenon of Ofar Bar-Yosef and others.

• Perhaps change “not as behaviorally distinct as once thought.” To “not as behaviorally distinct as previously considered.” Or something else—“once thought.” Is unclear.

This reviewer would like to thank the first author and Greg Mathews for assuaging my previous concerns about multiple groups in the values for Tfv and epLsar by doing the density plots with kernel density estimates to estimate probability and with histograms to clarify the results. The experiment demonstrates the excellent sample sizes allowing for a statistical treatment of data that showed patterns based on individual values as nonsigificant. The additional analysis improves my confidence in the validity of the results.

Reviewer #2: The authors have resubmitted a revised version of their manuscript, titled “Anterior tooth-use behaviors among early modern humans and Neandertals”, but unfortunately most of the reviewers’ comments were not taken into account.

In my opinion the response to reviewers included in the rebuttal letter is not really sufficient to address major criticisms of their work. Specifically, I find very odd to discuss about Neanderthal anterior tooth wear, culture and bite force without even mentioning the anterior dental loading hypothesis. The fact that in other Neanderthal anterior tooth-use studies there was no mention the biomechanics of Neanderthal anterior tooth wear, it is not a valid excuse to ignore this important aspect, which indirectly it is strictly associated with cultural habits in this human species.

Overall, I still feel that the manuscript is largely incomplete, in terms of background information, interpretation, discussion of the results and literature review. The authors also did not acknowledge any of the limitations of their study. While I agree with the authors that the sample size is good for a paleoanthropological study, it is never ideal from a statistical point of view. I think it is always worth being cautions when interpreting the results from the analysis of relatively small fossil samples. Generally, a sentence about the limitation of the study at the end of the discussion is sufficient.

Finally, as I have mentioned in my previous review, the authors did not highlight enough what is new and what is not new in their study.

7. PLOS authors have the option to publish the peer review history of their article (what does this mean?). If published, this will include your full peer review and any attached files.

Reviewer #1: No

Reviewer #2: No

---

## [Author Response · Author response to Decision Letter 1]

14 Oct 2019

Review of Krueger et al. for PONE: 

The authors have successfully responded to the critiques and suggestions of the reviewers. The result is an improved manuscript that is nearly ready for publication. The comments below are meant to further improve this excellent analysis and comparison of Middle and Upper Paleolithic anterior tooth texture. Although the relationship between anterior tooth use and cultural inferiority / superiority is not immediately obvious, in the Discussion the authors explain better how perceptions of Neandertals can influence the explanation of the variation in tooth wear and other studies. The lack of differences imply that similar behaviors were performed by Middle and Upper Paleolithic peoples and some of these were mimicked by their Holocene counterparts. The paper will have a high impact on the field and I encourage the rapid publication of the manuscript. 

Specific comments: 

Introduction 

• “had a more complex division of labor for resource acquisition” doesn’t quite make sense. More complex than whom? Maybe omit “more” “more” removed.

• “show distinctions between Neandertals and early modern humans – albeit, with some overlap” —consider revising, or perhaps replace with statistically meaningful terms such as “show a pattern in the distinctions in the mean values for Neandertal and early modern humans, but with large standard deviations resulting in nonsignificant differences…” We have revised this section to be more statistically meaningful. Please see lines 133-137. We also added in lines 137-145 about dentin exposure to help buttress the data about enamel wear.

M&M

• Table 2: Just double checking: you investigated La Quina 1, not La Quina 5, correct? In any event, it might be interesting to eventually examine the incisal microtexture of La Quina 5.

We investigated La Quina 5; we apologize for any confusion here. Please see the S1 supplementary data file for more information.

• “under the auspices of three attainable factors:” is unclear—consider changing to “using three factors” We changed this per the reviewer’s suggestion.

• Change “environmental reconstructions, dating techniques, etc” to “environmental reconstructions and dating techniques” We changed this per the reviewer’s suggestion.

• “[S1 file and SOM in 37; however, these limitations resulted in broad categories. We recognize that other researchers may use different groupings [101, 102].” Is problematic. The first clause is not a complete sentence yet includes a period and the end misses a double bracket. I recommend changing it to “as shown in the S1 file [and SOM in 37]; however, these limitations resulted in broad categories [cf. 101, 102].” We changed this per the reviewer’s suggestion.

• “Underpinnings” is unclear: Change “their mathematical underpinnings are described in Scott et al.” to “their mathematical descriptions are detailed in Scott et al.” We changed this per the reviewer’s suggestion.

• To increase clarity, maybe replace “time), six combinations in total, a” with “time)--six combinations in total—a” We changed this per the reviewer’s suggestion.

Results

• Just a recommendation—the authors might want to comb through the paper carefully and omit redundancy. For example, in the Results, I felt like l read that “Closed” habitat is removed in this analysis as no early modern humans analyzed here lived in “closed” environments.” or something like it multiple times in the text and captions—maybe remove one of them? There are a few others that are repeated once too often –maybe remove from table and figure captions? We changed this per the reviewer’s suggestion. Thank you for noticing the redundancy!

Discussion and conclusion

• I’d recommend removing entirely “While comparisons with the Neandertal sample will be addressed in more detail below, the similarity with the Point Hope Tigara commands consideration.” We removed this per the reviewer’s suggestion.

• Consider changing “in European Upper Paleolithic archaeological sites and those taxa used ethnographically for the fur, hide, sinew, and other raw materials” to “in European Upper Paleolithic archaeological sites and those taxa reported in the ethnographic record as sources of the fur, hide, sinew and other raw materials” We changed this per the reviewer’s suggestion.

• “Sadlermiuuit (Table 4).: still retains track-changes and later on lines 429, 430 We corrected these errors. 

• Lines 439-444--I think what we are seeing here in terms of the similarity of textures is the eat what you can find phenomenon of Ofar Bar-Yosef and others. Agreed! 

• Perhaps change “not as behaviorally distinct as once thought.” To “not as behaviorally distinct as previously considered.” Or something else—“once thought.” Is unclear. We changed this per the reviewer’s suggestion.

This reviewer would like to thank the first author and Greg Mathews for assuaging my previous concerns about multiple groups in the values for Tfv and epLsar by doing the density plots with kernel density estimates to estimate probability and with histograms to clarify the results. The experiment demonstrates the excellent sample sizes allowing for a statistical treatment of data that showed patterns based on individual values as nonsigificant. The additional analysis improves my confidence in the validity of the results.

We thank the reviewer for providing the idea in the first place. We were happy to examine the data further to see if anything significant was missed. 

Reviewer #2: The authors have resubmitted a revised version of their manuscript, titled “Anterior tooth-use behaviors among early modern humans and Neandertals”, but unfortunately most of the reviewers’ comments were not taken into account.

We did not add much of this reviewer’s comments in the manuscript; however, that does not mean the comments were neither meaningful nor impactful. Indeed, these comments forced us to think about our data using a biomechanical perspective, and if that framework should be added to the discussion at hand. We ultimately decided that adding in a review section outlining both biomechanical and comparative approaches was a better fit for the direction of this paper. 

In my opinion the response to reviewers included in the rebuttal letter is not really sufficient to address major criticisms of their work. Specifically, I find very odd to discuss about Neanderthal anterior tooth wear, culture and bite force without even mentioning the anterior dental loading hypothesis. The fact that in other Neanderthal anterior tooth-use studies there was no mention the biomechanics of Neanderthal anterior tooth wear, it is not a valid excuse to ignore this important aspect, which indirectly it is strictly associated with cultural habits in this human species.

We thoroughly addressed this reviewer’s concerns in the “response to reviewers” letter; however, we appreciate the time and effort this reviewer spent with this manuscript again. As such, we have included a new section in the Introduction titled “Biomechanical versus Comparative Approach,” which details the Anterior Dental Loading Hypothesis, challenges of such work, comparative approaches, and limitations of our work. We hope that this will reflect a good faith effort in showing Reviewer 2 that we have, indeed, chosen the best framework for our data. 

Moreover, a biomechanical approach focuses on the potential for Neandertals to produce (or modern humans to not produce) a high or heavy anterior bite force. Our data show that regardless of the potential for such behaviors, both hominins were actually performing these behaviors. These data do not provide any indication of whether the unique craniofacial traits found on Neandertals reflect an adaptation or a neutral evolutionary force, such as genetic drift, so to couch our discussion in such a framework is inappropriate. We understand the reviewer may not agree, but we are confident in this assessment of our data.

Overall, I still feel that the manuscript is largely incomplete, in terms of background information, interpretation, discussion of the results and literature review. The authors also did not acknowledge any of the limitations of their study. While I agree with the authors that the sample size is good for a paleoanthropological study, it is never ideal from a statistical point of view. I think it is always worth being cautions when interpreting the results from the analysis of relatively small fossil samples. Generally, a sentence about the limitation of the study at the end of the discussion is sufficient.

We added in limitations of direct, comparative approaches, including our study, in lines 187-189. 

Finally, as I have mentioned in my previous review, the authors did not highlight enough what is new and what is not new in their study.

We respectively disagree with this assessment once again. There has never been a study that analyzes dental microwear as objectively as the method used here. Moreover, the bioarchaeological baseline used here provides a comprehensive framework for interpreting the actual behaviors of early modern humans versus those of their Neandertal counterparts. This study offers an innovative and unique lens into the lives of these hominins.

---

## [Editor Report · Decision Letter 2]

17 Oct 2019

Anterior tooth-use behaviors among early modern humans and Neandertals

PONE-D-19-17387R2

Dear Dr Krueger,

We are pleased to inform you that your manuscript has been judged scientifically suitable for publication and will be formally accepted for publication once it complies with all outstanding technical requirements.

With kind regards,

Ron Pinhasi

Academic Editor

PLOS ONE
---

## [Editor Report · Acceptance letter]

24 Oct 2019

PONE-D-19-17387R2 

Anterior tooth-use behaviors among early modern humans and Neandertals 

Dear Dr. Krueger:

I am pleased to inform you that your manuscript has been deemed suitable for publication in PLOS ONE. Congratulations! Your manuscript is now with our production department. 

With kind regards,

on behalf of

Dr. Ron Pinhasi 

Academic Editor

PLOS ONE